# The Arctic Ocean Observation Operator for 6.9 GHz (ARC3O) - Part 1: How to obtain sea-ice brightness temperatures at 6.9 GHz from climate model output

Clara Burgard[1,2], Dirk Notz[1,3], Leif T. Pedersen[4], and Rasmus T. Tonboe[5]

[1]Max Planck Institute for Meteorology, Hamburg, Germany
[2]International Max Planck Research School for Earth System Modelling, Hamburg, Germany
[3]Institute of Oceanography, Center for Earth System Research and Sustainability, Universität Hamburg, Hamburg, Germany
[4]National Space Institute, Technical University of Denmark, Lyngby, Denmark
[5]Danish Meteorological Institute, Copenhagen, Denmark

**Correspondence:** Clara Burgard (clara.burgard@mpimet.mpg.de)

**Abstract.** We explore the feasibility of an observation operator producing passive microwave brightness temperatures for sea ice at a frequency of 6.9 GHz. We investigate the influence of simplifying assumptions for the representation of sea-ice vertical properties on the simulation of microwave brightness temperatures. We do so in a one-dimensional setup, using a complex 1D thermodynamic sea-ice model and a 1D microwave emission model. We find that realistic brightness temperatures can be simulated in cold conditions from a simplified linear temperature profile and a simplified salinity profile as a function of depth in the ice. These realistic brightness temperatures can be obtained based on profiles interpolated to as few as five layers. Most of the uncertainty resulting from the simplifications is introduced by the simplification of the salinity profiles. In warm conditions, the simplified salinity profiles lead to too high brine volume fractions in the subsurface layer. To overcome this limitation, we suggest using a constant brightness temperature for the ice during warm conditions and to treat melt ponds as water surfaces. Finally, in our setup, we cannot assess the effect of wet snow properties. As periods of snow with intermediate moisture content, typically occuring in spring and fall, locally last for less than a month, our approach allows one to estimate realistic brightness temperatures at 6.9 GHz from climate model output for most of the year.

## 1 Introduction

Sea-ice concentration products are retrieved from passive microwave brightness temperatures measured by satellites and come with a non-negligible uncertainty (Ivanova et al., 2015; Tonboe et al., 2016; Lavergne et al., 2019). This observational uncertainty hinders reliable climate model initialization (Bunzel et al., 2016) and model evaluation (Notz et al., 2013). Additionally, it hinders a robust extrapolation of the future sea-ice evolution based on current observations. For example, sea-ice area is strongly coupled to changes in the global-mean air temperature (Gregory et al., 2002; Winton, 2011; Mahlstein and Knutti, 2012; Ridley et al., 2012; Li et al., 2013) and thus to $CO_2$ emissions (Notz and Stroeve, 2016). The relationship between $CO_2$ emissions, global-mean air temperature and sea ice provides the possibility to project the future Arctic sea-ice evolution under different forcing scenarios. However, Niederdrenk and Notz (2018) showed that the observational uncertainty in sea-ice

concentration translates into uncertainty in the sensitivity of sea ice to changes in global-mean air temperature and therefore leads to uncertainty in the temperature at which an ice-free Arctic in summer can be expected.

Observation operators are a current approach in climate science to circumvent observational uncertainty and the spread introduced by the use of retrieval algorithms on satellite measurements (Flato et al., 2013; Eyring et al., 2019). They simulate directly the observable quantity, in our case the brightness temperature, from the climate model output instead of retrieving the simulated quantity, in our case the sea-ice concentration, from the satellite observations. A sea-ice observation operator reduces the uncertainty introduced by assumptions used in retrieval algorithms about the state of other climatic variables besides the sea-ice concentration. It takes advantage of knowing the consistent climate state in time and space simulated by the climate model alongside the sea ice. This knowledge allows a more comprehensive approach to climate model evaluation, as we cannot only assess the simulated sea-ice concentration but also the simulated sea-ice temperature, snow cover, and sea-ice type. The feasibility and limitations of an observation operator applied to sea ice simulated by a climate model have not been investigated yet. This is the question we address here.

We investigate how important the complexity of the representation of sea-ice properties is for the simulation of sea-ice surface brightness temperatures emitted by different ice types. Experiments using a model accounting for part of the processes at work inside the sea ice combined with an emission model have shown that knowing the vertical sea-ice properties are sufficient to generate realistic microwave brightness temperatures (Tonboe, 2010; Tonboe et al., 2011). We mainly concentrate on the vertical representation of temperature and salinity inside the ice and snow, as they are the main drivers of the brine volume fraction in the ice and liquid water fraction in the snow and thus of sea-ice brightness temperatures, especially at low microwave frequencies (Ulaby et al., 1986). As most general circulation models (GCMs) do not explicitly represent the time evolution of vertical profiles of temperature and salinity in the ice and snow, we investigate the effect of simplified temperature and salinity profiles on the simulation of brightness temperatures. We do so by comparing reference profiles, representing an estimate of reality, on the one hand and simplified profiles, representing GCM output, on the other hand in an idealized one-dimensional setup, using a complex thermodynamic sea-ice model and a microwave emission model.

We focus on the simulation of sea-ice brightness temperatures at 6.9 GHz at vertical polarization as a first step. At this frequency, the main driver of brightness temperatures are the sea-ice properties, while the contribution of the snow and of the atmosphere due to water vapor, cloud liquid water and temperature are small compared to the surface contribution. The framework can, however, be extended to other frequencies and polarizations in the future, if the increasing importance of the snow and atmospheric contribution with increasing frequency is taken into account.

In Sec. 2, we provide the theoretical background about drivers of sea-ice brightness temperatures and in Sec. 3 we present our method and the sea-ice and emission models used for our experiments. In Sec. 4, we explore the influence of simplifications in the temperature and salinity profiles on the simulation of sea-ice brightness temperatures to then explore the effect of a reduced number of layers. Finally, we discuss our results in Sec. 5 and conclude with suggestions for a functional observation operator for sea ice in Sec. 6.

## 2 Theoretical background

The brightness temperature is a measure for the microwave radiation emitted by one medium or a combination of media and corresponds to the temperature of a blackbody emitting the observed amount of radiation. It depends on the temperature distribution in the medium and on the transmission and reflection affecting the path of the microwave radiation from the emitting layer within the medium to the surface of the medium. The transmission and reflection in turn depend on the properties of the medium and on the frequency and polarization of the radiation.

Transmission and reflection of the microwave radiation within an ice column are driven by the permittivity and the dielectric loss of the different layers of the ice on the one hand and scatterers present in the ice on the other hand. Sea ice is a mixture of liquid brine and pure ice and the permittivity and dielectric loss of liquid brine are orders of magnitude larger than the permittivity and dielectric loss of pure ice (Ulaby et al., 1986; Shokr and Sinha, 2015b). Therefore, the permittivity and dielectric loss inside a sea-ice column are mainly a function of the fraction and distribution of liquid brine in the different layers of the ice. This means that, looking at a vertical profile of the ice, ice layers with high brine volume fractions have a lower transmissivity and larger reflectivity than ice layers with low brine volume fractions. The vertical distribution of the brine volume fraction in the ice is a function of the vertical distribution of temperature and salinity. Brine is present within the ice throughout its first year. If the ice becomes multiyear ice, most of its brine will have drained out and the brine volume fraction decreases substantially compared to first-year ice.

The scattering within an ice column is a function of the permittivity and the size of scatterers inside the ice. In first-year ice, the main scatterers are brine pockets, while in multiyear ice the main scatterers are air bubbles, as most of the brine will have drained out (Winebrenner et al., 1992; Tonboe et al., 2006; Shokr and Sinha, 2015a).

As brightness temperatures are usually not measured at the ice surface but at the top of the atmosphere by satellites, the microwave radiation emitted by the sea-ice cover can additionally be affected by transmissivity and reflectivity of the snow and atmosphere on the path between the surface and the satellite. For frequencies below 10 GHz, dry snow is practically "transparent" (Hallikainen, 1989) and the atmosphere has a negligible influence. For frequencies higher than 10 GHz, scattering occurs within a dry snowpack (Mätzler, 1987; Barber et al., 1998). In general, scattering affects the brightness temperature measured from space over sea-ice surfaces increasingly with increasing frequency (Tonboe et al., 2006) as the wavelength successively approaches the size of brine pockets and air bubbles on the order of tenths of millimeters to millimeters, snow grains on the order of hundreds of micrometers to millimeters and atmospheric aerosols and droplets on the order of micrometers.

If the snow becomes wet, as happens during melting periods and localized events of warm air advection mainly occurring in spring and fall, the dielectric loss in the snow layers increases substantially, leading to a reduction in the transmissivity of the snow layer to microwave radiation. This may also happen when brine wicking takes place in the lowest layer of the snow, especially above first-year ice (Barber et al., 1998; Shokr and Sinha, 2015b). However, we will not attempt to investigate in detail the effect of wet snow on the radiation in this study as our model setup does not allow us to simulate detailed processes within the snowpack.

Sea-ice concentration retrievals are based on satellite measurements at frequencies ranging from 1.4 GHz to 91 GHz (Ivanova et al., 2014, 2015; Gabarro et al., 2017). In the following, we concentrate on radiation at 6.9 GHz and vertical polarization. This frequency is advantageous as, with a wavelength of approx. 4.3 cm, it is only slightly affected by scattering inside the ice, the snow, and the atmosphere. The brightness temperature at 6.9 GHz therefore mainly depends on the properties affecting permittivity and dielectric loss of the different layers inside the ice. This is why our focus lies on the properties of the sea-ice column, rather than on the snow structure or the state of the atmosphere. The penetration depth in ice at 6.9 GHz is around 20 cm for first-year ice and around 50 cm for multiyear ice (Tonboe et al., 2006). Therefore, we investigate not only the properties of the ice surface but also the properties of the whole sea-ice column to be sure to capture the main influences on the brightness temperature.

## 3   Methods and Data

Although a few GCMs use detailed sea-ice modules (Vancoppenolle et al., 2009; Bailey et al., 2018), most GCMs use very simple sea-ice models that do not resolve the properties driving microwave transmission and reflection inside the ice and snow. Ideally, our observation operator would compute brightness temperatures from such a GCM as well. However, it is not clear yet how these simplifications affect a brightness temperature simulated based on a simple representation of the relevant properties. As a basis to investigate the effect of using non-detailed sea-ice information, we assume that our input for the operator would be output by the Max Planck Institute Earth System Model (MPI-ESM, Wetzel et al., 2012). In MPI-ESM, sea ice is represented as flat sea ice, with very simple sea-ice properties: a sea-ice (bare ice) or snow (snow-covered ice) surface temperature, a constant sea-ice bottom temperature at -1.8 °C, and a constant salinity of 5 g/kg regardless of sea-ice type or age (Notz et al., 2013).

To explore the importance of the vertical distribution of sea-ice properties on the simulation of brightness temperatures, we use an idealized one-dimensional setup. This one-dimensional setup works as follows. On the one hand, we use a one-dimensional thermodynamic sea-ice model to simulate our reference profiles (see Sec. 3.1). It computes highly resolved vertical sea-ice profiles under a given atmospheric forcing. On the other hand, we simplify these reference profiles to emulate profiles that could be inferred from information given by MPI-ESM for the same conditions. These two sets of profiles can be used to simulate two sets of brightness temperatures with a microwave emission model (see Sec. 3.2). The two sets of resulting brightness temperatures can then be used to quantify the effect of the GCM simplification on the brightness temperature simulation, compared to our reference (see Fig. 1, Sec. 3.3 and Sec. 3.4).

In this setup, we can quantify the influence of each parameter separately on the simulated brightness temperature. This a necessary first step to understand fundamental drivers of the brightness temperature before comparing brightness temperatures simulated on the basis of MPI-ESM output directly to brightness temperatures measured by satellites, which we do in Burgard et al. (2020).

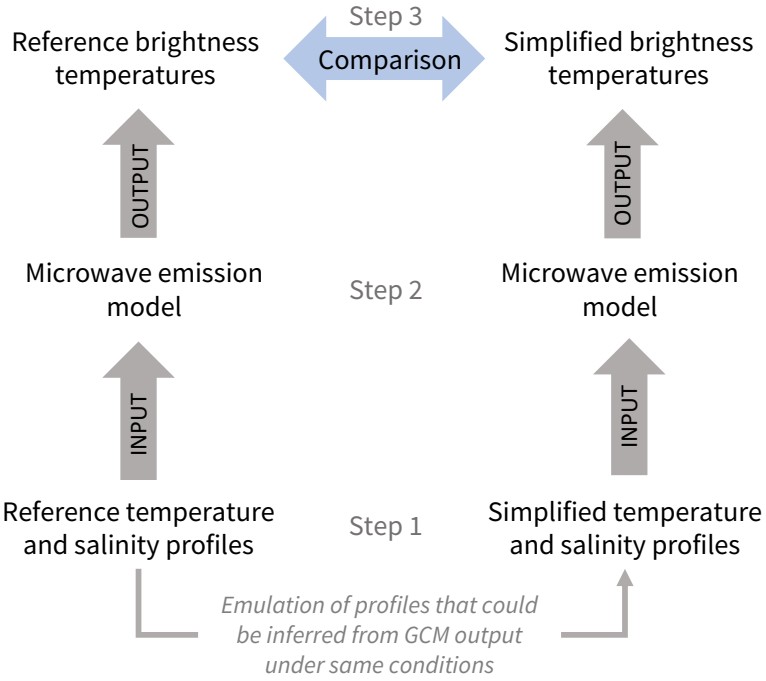

**Figure 1.** Schematic of the steps of our simulation and comparison method.

### 3.1 SAMSIM

Our reference profiles are simulated by the 1D Semi-Adaptive Multi-phase Sea-Ice Model (SAMSIM, Griewank and Notz, 2013, 2015). This is a complex thermodynamical model simulating the evolution of a 1D sea-ice column under given surface forcing. It computes sea-ice temperature, salinity, and brine volume fraction profiles on a semi-adaptive grid, with a number

of layers varying between 0 and 100. It includes most of the processes governing sea-ice growth and melt, and interactions between the ice and, if existent, its snow cover. It was developed to investigate the brine dynamics inside the ice. A detailed description of underlying equations and represented processes can be found in Griewank and Notz (2013) and Griewank and Notz (2015).

We force SAMSIM with 2 m air temperature, surface downward longwave radiation, surface downward shortwave radiation,

and precipitation from the ERA-Interim reanalysis (Dee et al., 2011) in the time period from July 2005 to December 2009. This gives us insight into 4.5 annual cycles, so that we can assess the interannual variability of the growth and melt of sea ice and the evolution of its properties. The ocean salinity is kept at 34 g/kg and the oceanic heat flux at the bottom of the ice is derived from SHEBA measurements, varying between 0 W/m$^2$ in spring and 14 W/m$^2$ in autumn (Huwald et al., 2005; Griewank and Notz, 2015).

We conduct our analysis using atmospheric forcing from two random points in the Arctic Ocean as input for SAMSIM. At the first point, the combined forcing of the ERA-Interim atmospheric variables and the SHEBA oceanic flux leads to

complete melting of the simulated ice in summer each year, resulting in several cycles of first-year ice. At the second point, the combination of the atmospheric forcing and oceanic heat flux leads to a simulated ice cover present throughout the year, resulting in several cycles of multiyear ice (Fig. 2). This way, we capture potential differences in the brightness temperature simulation depending on the ice type. To ensure that the conclusions we draw from these two random points are robust, we

5   have conducted the same analysis on five additional random points distributed in the Arctic Ocean and the results support our conclusions.

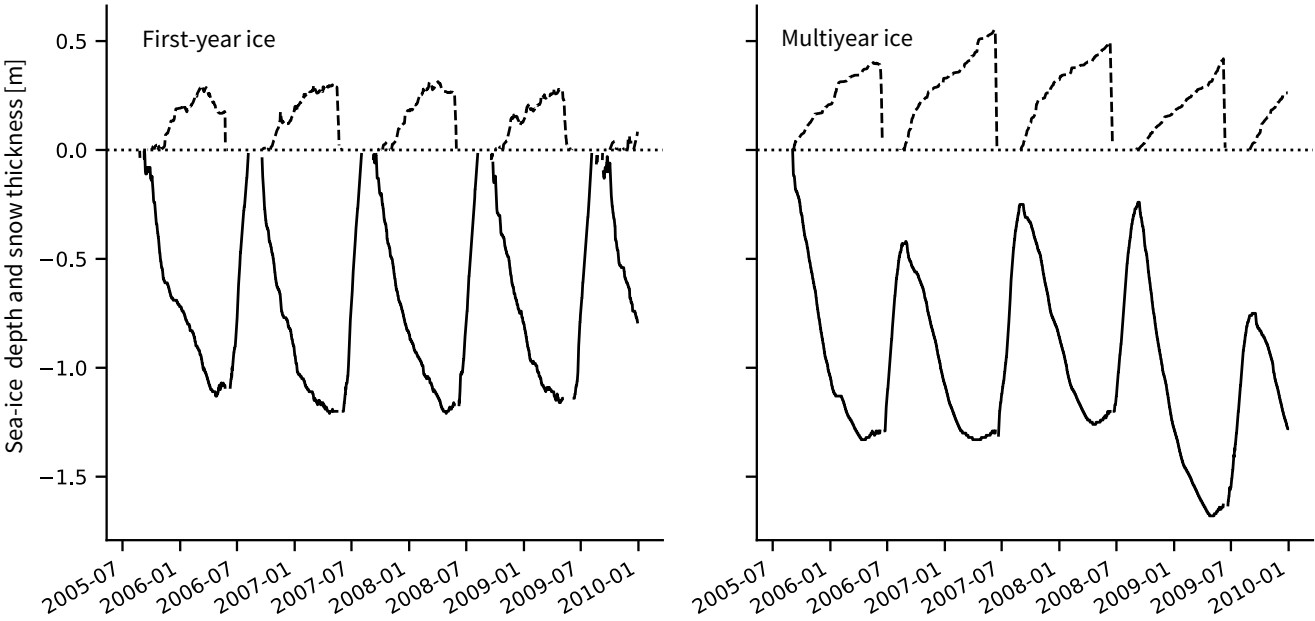

**Figure 2.** Evolution of sea-ice (full line) and snow (dashed line) thickness as simulated by SAMSIM under ERA-Interim forcing between July 2005 and December 2009. We use ERA-Interim forcing from 75 °N00 °W for the first-year ice and from 90 °N for multiyear ice. Note that, to avoid unrealistic model artifacts in the ice thickness, we have masked out the few timesteps following the final phase of the melting of the snow cover. Also note that the same analysis was conducted using atmospheric forcing from the points 74 °N170 °E, 77 °N39 °E, 80 °N160 °W, 82 °N120 °W, 85 °N50 °W (not shown) to ensure the robustness of our results.

### 3.2   MEMLS

The simulation of sea-ice brightness temperatures is conducted with a slightly modified version of the Microwave Emission Model for Layered Snowpacks (MEMLS) extended to sea ice (Tonboe et al., 2006). MEMLS was first developed by Wiesmann

10   and Mätzler (1999) to simulate brightness temperatures emitted by a snowpack composed of several layers and was later extended to sea ice (Tonboe et al., 2006). MEMLS uses the information of the properties of the ice and snow layers to simulate the path of microwave radiation from the bottom to the surface of the ice and, if present, snow. It uses the thickness, the

temperature, the salinity, the density, the correlation length (measure for the scatterer size), the wetness, the brine pocket form, and information about the type of medium (snow, first-year/multiyear ice) of the different sea-ice and snow layers to compute transmission and reflection of the radiation along the path. This then results in a brightness temperature emitted at the surface of the ice or snow.

We do not take into account the atmosphere in our analysis as its effect is relatively small at 6.9 GHz. The use of the term "brightness temperatures" in the following is therefore equivalent to the use of "brightness temperatures emitted at the surface of the ice and snow column".

## 3.3   General simulation setup

The temperature and salinity profiles produced by SAMSIM are used as input for MEMLS for the simulation of brightness
temperatures. Additionally, density profiles are derived from temperature and salinity using relationships given by Notz (2005) (see Eq. A5). Next to the temperature, salinity and density profiles, other variables, which are not computed by SAMSIM, have to be provided to MEMLS. These are the correlation length, the brine pocket form, the incidence angle, the ocean temperature, the incoming microwave radiation from the atmosphere (i.e. the cosmic background radiation and the radiation reflected and emitted by properties of the atmosphere) and the ice-ocean reflectivity for vertical polarization. They are set to constants, listed
in Table  1.

Additionally, except for snow thickness and temperature, snow properties are neither resolved in SAMSIM nor in MPI-ESM. Although a dry snow cover is practically "transparent" at frequencies lower than 10 GHz (Hallikainen, 1989), we still need to account for its presence due to one indirect and one direct effect on the brightness temperature. On the one hand, the snow cover leads to the thermal insulation of the ice column and therefore affects the temperature profile inside the ice, which in
turn affects the brightness temperature. On the other hand, the difference in density between ice, snow and atmosphere leads to refraction of the radiation at the interface between ice and snow and between snow and atmosphere. The former effect is taken into account through the use of the SAMSIM snow thickness and snow temperature evolution, and the latter is taken into account through the snow thickness and by using a low density for snow compared to ice. We therefore set all snow properties, except the snow temperature and snow thickness, to constants, also listed in Tab. 1.

The effect of wet snow on the brightness temperature is larger and depends on the snow wetness, brine wicking, and snow metamorphism. As neither SAMSIM or MPI-ESM resolve these properties in the snow, we set the snow wetness to zero in this idealized study. However, when comparing results of a possible observation operator based on this study to actual observations, we strongly recommend to not consider periods of wet snow, during melting periods and events of warm air advection, as setting the snow wetness to zero will lead to unplausible brightness temperatures in these periods.

Our input for the emission model, e.g. salinity, correlation length, brine pocket form, comes with uncertainties. These are mainly caused by a partial or complete lack of in-situ observations of these small-scale properties and the resulting low understanding of their evolution. We therefore recommend more observations of the ice properties, ideally combined with concurrent microwave radiation measurements. A few of such observations exist already, from both laboratory setting and in-situ, but they mainly focus on frequencies higher than 6.9 GHz (e.g. Grenfell et al., 1998; Jezek et al., 1998; Perovich et al., 1998; Hwang

**Table 1.** MEMLS constant input details and properties of the snow layer. The incidence angle is from AMSR-E and AMSR2, passive microwave sensors measuring at 6.9 GHz (NASDA, 2003; JAXA, 2011). The ocean temperature and snow density are the constant values used in a GCM such as MPI-ESM (Wetzel et al., 2012; Giorgetta et al., 2013). The incoming microwave radiation from the atmosphere is set to 0 K because we want to focus on the effect of sea-ice properties on the emitted radiation. Correlation lengths are based on past experiments conducted by R.T. Tonboe.

| | |
|---|---|
| Incidence angle | 55° |
| Ocean temperature | -1.8 °C |
| Incoming microwave radiation from the atmosphere | 0 K |
| Ice-ocean reflectivity for V-polarization | 0.25 |
| Brine pocket form | spherical |
| Correlation length first-year ice | 0.35 mm for depth $< 20$ cm, 0.25 mm for depth $> 20$ cm |
| Correlation length multiyear ice | 1.5 mm |
| Snow thickness | as computed by SAMSIM |
| Snow density | 300 kg/m$^3$ |
| Snow correlation length | 0.15 mm |
| Snow salinity | 0 g/kg |
| Snow temperature | as computed by SAMSIM |

et al., 2007). With more combined observations at lower frequencies, we expect that the uncertainty in the brightness temperature simulation can be reduced in the future through further research and better understanding of the components introducing the uncertainty.

For example, a better understanding of the sea-ice salinity evolution would be of advantage. The salinity parametrization used in Sec. 4.2.2 is based on an "L-shape" of the salinity profile, while the sea-ice salinity profile often resembles a "C-shape" or even a "Γ-shape" when cold temperatures prevail (Nakawo and Sinha, 1981; Shokr and Sinha, 2015a). Another parameter of uncertainty is the correlation length. Although it is a variable quite well understood and quantifiable for snow (Mätzler, 2002; Proksch et al., 2015; Lemmetyinen et al., 2018), its quantification in sea ice is not clear and its values not well known. On a similar note, MEMLS uses assumptions about the form of the brine pockets. Here, we assume spherical brine pockets. However, it is known that the brine pocket form highly depends on the initial formation process of the ice, which is not simulated. In any case, we assume that the choice of brine pocket form will not affect our result substantially because scattering within the ice is negligible at 6.9 GHz.

Another limitation in the input data for MEMLS is the snow information. We investigated the indirect effect of the snow cover on the simulated brightness temperature, e.g. the radiative effect (as opposed to the thermal insulation effect), and found that the brightness temperature decreases by approximately 0.13 K for every centimeter of snow present on the ice column.

Therefore, although the snow is expected to be "transparent" at less than 10 GHz, lack of information about the snow structure besides snow temperature and thickness might still lead to uncertainties of up to a few K in the presence of a thick snow cover.

Finally, the use of MEMLS as a sea-ice emission model is a source of uncertainty as well. Here again, the lack of measurements of the parameters needed for the brightness temperature simulation and of microwave radiation itself has inhibited a comprehensive evaluation of the sea-ice version of MEMLS simulations against reality. Still, it is accepted as one of the main tools for sea-ice brightness temperature simulations and has shown its strength in several previous studies (Tonboe, 2010; Tonboe et al., 2011; Willmes et al., 2014; Lee et al., 2017).

However, the uncertainties listed above only have a limited impact on the present study. We concentrate on a relative comparison, where we change temperature and salinity in the ice to understand their impact on the brightness temperature, but assumptions about the snow and ice correlation length, the form of brine pockets, and the snow density are the same in our reference and our simplified brightness temperature simulations. The uncertainties will therefore not impact the difference between the two sets of brightness temperatures. Additionally, in regard to the absolute values, Burgard et al. (2020) show that realistic brightness temperatures can be simulated by MEMLS using the above mentioned uncertain assumptions with slight tuning. The effect of the uncertainties therefore remains small when considering large scales.

## 3.4  Experiments

The aim of this study is to assess if realistic brightness temperatures can be simulated for 6.9 GHz, vertical polarization, using the limited information about sea-ice properties provided by a GCM such as MPI-ESM. This assessment is conducted through a range of experiments. In a first step (see Sec. 4.1), we investigate the influence of the ice surface and subsurface properties on the radiation emitted by the snow-ice column. We examine in which conditions information about the vertical profile is needed for realistic brightness temperatures to be simulated and in which conditions information about surface and subsurface properties is enough.

In a second step (see Sec. 4.2), we examine the effect of assuming a linear temperature profile and of different assumptions for the simplification of the salinity profile on the simulated brightness temperature. In this set of experiments, we compare brightness temperatures simulated based on SAMSIM profiles (in the following our *reference profiles*) and brightness temperatures simulated based on the *simplified profiles*. The simplified input profiles are interpolated to the same number of layers as the reference profiles (ranging from 1 to 100 layers, depending on the ice thickness).

In a third step (see Sec. 4.3), we examine the effect of reducing the vertical resolution on the simulated brightness temperature. To do so, we interpolate the vertical properties on fewer layers than the reference profiles.

## 4 Results

### 4.1 Subsurface properties vs. Vertical profile

Sea-ice brightness temperatures at 6.9 GHz are mainly driven by the distribution of liquid brine inside the ice, as the permittivity and dielectric loss of the ice layers play a larger role than scattering at this frequency. We compute the brine volume fraction with Eq. A4 based on the ice temperature and salinity profiles given by SAMSIM. Comparing the ice subsurface brine volume fraction, i.e. in the top ice layer (upper one centimeter) of the profiles, with the simulated reference brightness temperatures, the relationship between brine volume fraction and brightness temperature is clearly visible. The brightness temperatures show a strong dependence on the ice subsurface brine volume fraction (Fig. 3, top row). If we concentrate the brightness temperature simulation on the ice layers, i.e. using only the properties of the ice layers of the snow and ice column as input to MEMLS, the slight offset in the brightness temperature introduced by the refraction due to the snow cover is removed and the relationship is even clearer (Fig. 3, bottom row).

When the ice subsurface brine volume fraction is higher than 0.2, the brightness temperature from the ice column is linearly related to the ice subsurface brine volume fraction (Fig. 3, bottom row). This means that no radiation signal from below the subsurface layer influences the brightness temperature and only the brine volume fraction in the upper centimeters of ice matters. The brightness temperature varies roughly linearly between brightness temperatures typical for ice ($\approx 260$ K) at an ice subsurface brine volume fraction of 0.2 and brightness temperatures typical for open water ($\approx 160$ K) at an ice subsurface brine volume fraction of 1. In our SAMSIM profiles, these high subsurface brine volume fractions occur predominantly in warm conditions, i.e. from April to September, during the melting period and in the beginning of the freezing season. We therefore suggest that an ice subsurface brine volume fraction above 0.2 can be interpreted both as very wet ice or as a measure for the melt-pond fraction. This strong relationship means that, when the brine volume fraction is above 0.2, the subsurface properties play the main role for the brightness temperature simulation and vertical properties are not necessarily needed.

In some multiyear ice cases during warm conditions, the brightness temperature drops below 240 K at near-zero subsurface brine volume fractions. These low brightness temperatures occur in September, in the first two or three weeks in which ice growth sets in again. In these cases, the ice column used as input for MEMLS has a brine volume fraction of zero over the whole column, except in the bottom layer. We therefore suggest that the simulated brightness temperature is mainly influenced by the very saline bottom layer at the interface between ice and ocean in these cases, leading to low brightness temperatures. This behaviour is not necessarily realistic and the conditions leading to these input salinity profiles might need further investigation.

Otherwise, for subsurface brine volume fractions below 0.2, occurring in both cold and warm conditions, the brightness temperatures vary by 10 to 15 K around 260 K for similar ice subsurface brine volume fractions. For these low ice subsurface brine volume fractions, the brightness temperatures are driven by the distribution of brine further inside the ice, which is a function of the temperature and salinity distribution. Unfortunately, for these brightness temperatures around 260 K at low ice subsurface brine volume fractions, we could not infer a direct relationship between the brightness temperature and a given layer or a given brine volume fraction inside the ice from our data. This implies that information about the vertical distribution of

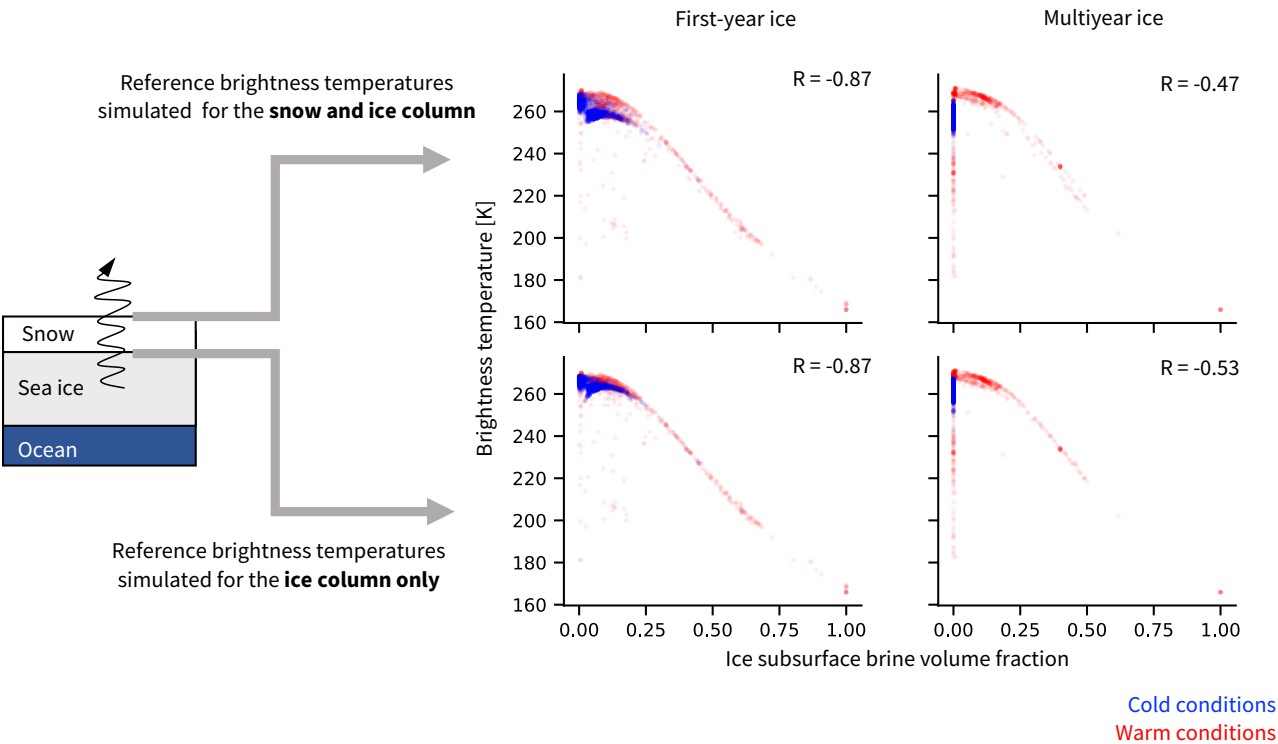

**Figure 3.** Reference brightness temperatures at 6.9 GHz, vertical polarization, simulated based on properties simulated by SAMSIM for the ice and snow column (top row) and on the ice column only (bottom row) as a function of the reference ice subsurface brine volume fraction for first year-ice (left column) and multiyear ice (right column). Blue is cold conditions (October to March), red is warm conditions (April to September). $R$ is the correlation coefficient between the brightness temperature and the ice subsurface brine volume fraction.

temperature and salinity, and consequently brine volume fraction, throughout the ice column is necessary to simulate realistic brightness temperatures.

From this first look at the relationship between ice properties and simulated brightness temperatures, we conclude that information about the vertical profiles of brine volume fraction are necessary for the simulation of brightness temperatures

5  for cold conditions and for parts of the warm conditions. The effect of describing the brine volume fraction profiles through simplified temperature and salinity profiles on the brightness temperature simulation is what we investigate in a next step.

## 4.2  Simplifying the temperature and salinity profile

The brightness temperature emitted by a snow and ice column is mainly driven by the distribution of the brine volume fraction in the ice column. As the brine volume fraction can be described as a function of temperature and salinity, we now investigate

10  the effect of reduced information availability about these profiles, as would be the case in GCM output, on the simulated brightness temperatures.

### 4.2.1 Simplifying the temperature profile

We start by investigating the brightness temperature simulated based on a temperature profile as could be inferred from MPI-ESM output. We call this experiment SIMPLETEMP. MPI-ESM computes a sea-ice (bare ice) or snow (snow-covered ice) surface temperature and a constant sea-ice bottom temperature at -1.8 °C. Therefore, we suggest using a two-step linear profile through snow and ice. We use the snow surface temperature as simulated by SAMSIM and infer the ice temperature at the interface between ice and snow from it, following Eq. A6. From this ice surface temperature, we interpolate the temperature profile linearly to the ice bottom layer, which has a temperature of -1.8 °C.

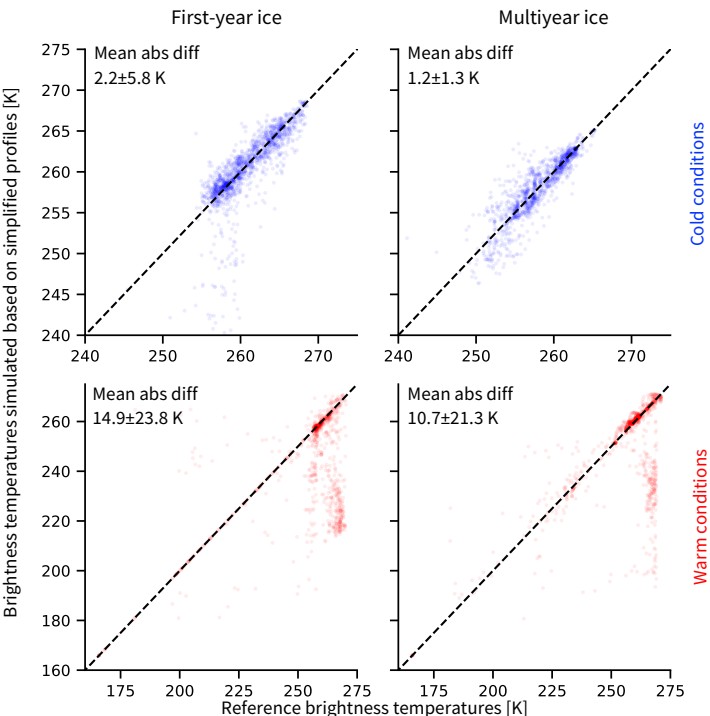

**Figure 4.** Brightness temperatures at 6.9 GHz, vertical polarization, simulated based on linear temperature profiles and reference salinity profiles (experiment SIMPLETEMP) as a function of reference brightness temperatures. Left column: first-year ice, right column: multiyear ice. Top row: Cold conditions (October to March), bottom row: Warm conditions (April to September). Note that the axes for cold conditions are limited to the range between 240 to 275 K for clarity. The remaining brightness temperatures are scattered between 165 and 240 K and represent around 2% of the simplified data and 0.4% of the reference data.

The influence of the simplifications is clearly different depending on the season. We therefore divide our results into cold conditions (October to March, see Fig. 4, top row) and warm conditions (April to September, see Fig. 4, bottom row). In

cold conditions, the absolute difference between brightness temperatures simulated from simplified profiles and brightness temperatures simulated from reference profiles remains small for both first-year ice (2.2±5.8 K) and multiyear ice (1.2±1.3 K). In warm conditions, this absolute difference increases by approximately one order of magnitude to 14.9±23.8 K (first-year ice) and 10.7±21.3 K (multiyear ice). The assumption of a two-step linear temperature profile in the snow and ice does therefore not introduce large uncertainties in the brightness temperature simulation in cold conditions but should be used with care in warm conditions.

### 4.2.2 Simplifying the salinity profile

In the experiment SIMPLESALCONST, we explore the effect of a constant salinity profile on the simulated brightness temperature. MPI-ESM assumes a constant salinity of 5 g/kg regardless of sea-ice type or age. As this is clearly too high for multiyear ice (Ulaby et al., 1986), we assume a constant salinity of 5 g/kg for first-year ice and a constant salinity of 1 g/kg for multiyear ice throughout the ice column in our simplified salinity profiles (see dashed lines in Fig. 5).

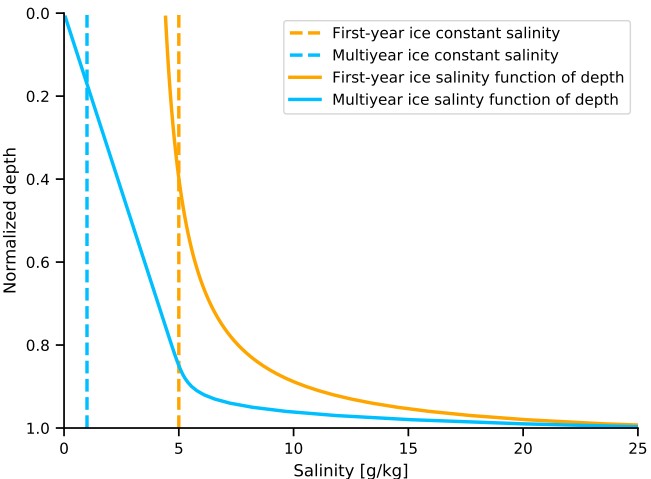

**Figure 5.** Salinity profiles used for the simplified profiles in Sec. 4.2.2. The dashed lines represent the constant salinity profiles and the full lines represent the salinity profiles as a function of depth. The colours represent the different ice types.

In the parallel experiment SIMPLESALFUNC, we explore an alternative approach to simplify salinity profiles. We use a parametrization representing salinity as a function of depth (Griewank and Notz, 2015). This parametrization assumes an L-shaped profile, with low salinity near the surface and a rapidly increasing salinity in the lower ice layers (see Fig. 5, full lines, and Table B1). This parametrization has been evaluated against observations (Griewank and Notz, 2015). In both SIMPLE-SALCONST and SIMPLESALFUNC, we use the reference temperature profiles simulated by SAMSIM.

Again, we divide the results depending on the season. While, for first-year in cold conditions, the effect of using a constant salinity (SIMPLESALCONST) is as low as using a linear temperature profile, with an absolute difference between the brightness temperatures based on simplified profiles and the reference brightness temperature of 2.5±6.5 K, the absolute difference

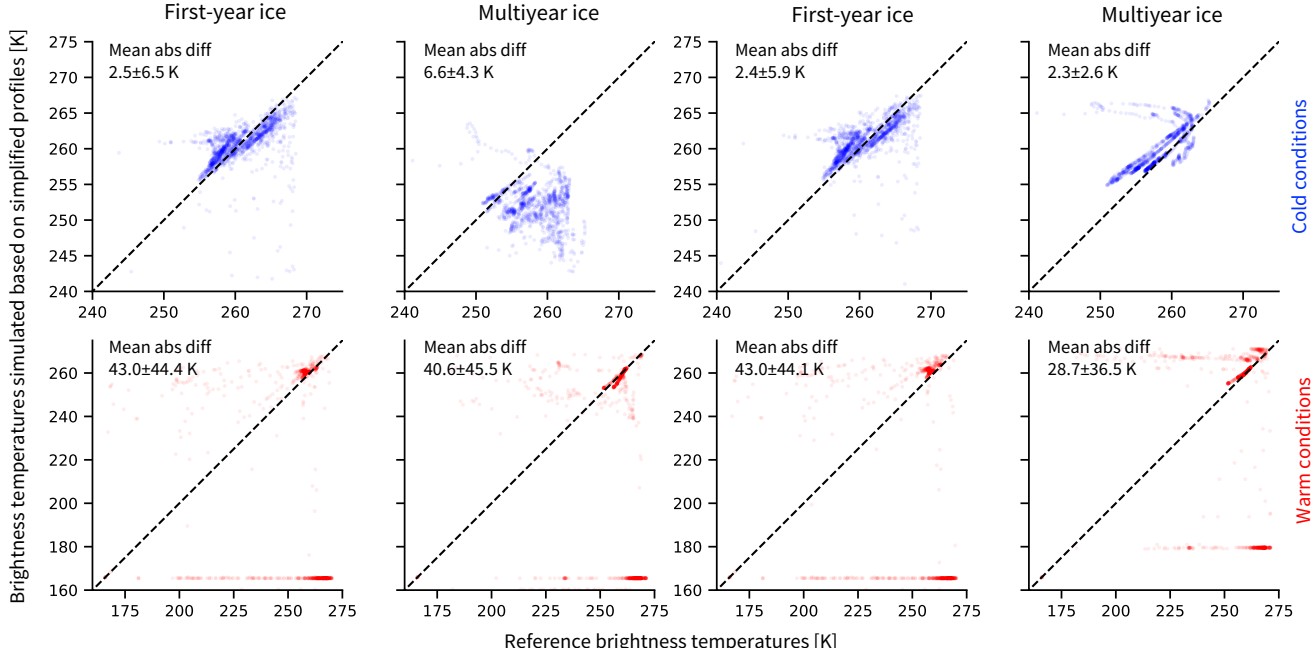

**Figure 6.** Brightness temperatures at 6.9 GHz, vertical polarization, simulated based on reference temperature profiles and (a) constant salinity profiles (experiment SIMPLESALCONST) or (b) salinity profiles as a function of depth (experiment SIMPLESALFUNC) as a function of reference brightness temperatures. 1st and 3rd column: first-year ice, 2nd and 4th column column: multiyear ice. Upper row: Cold conditions (October to March), lower row: Warm conditions (April to September). Note that the axes for cold conditions are limited to the range between 240 to 275 K for clarity. The remaining brightness temperatures are scattered between 165 and 240 K and represent around 2% of the simplified data and 0.4% of the reference data.

reaches 6.6±4.3 K for multiyear ice (Fig. 6a, top row). In warm conditions, the mean absolute differences are one order of magnitude higher, 43.0±44.4 K for first-year ice and 40.6±45.5 K for multiyear ice (Fig. 6a, bottom row).

If the brightness temperature is simulated based on reference temperature profiles and on the salinity profiles as a function of depth (SIMPLESALFUNC, Fig. 6b), the uncertainty is similar to the uncertainty introduced by using a constant salinity profile for first-year ice (2.4±5.9 K in cold conditions and 43.0±44.1 K in warm conditions). However, for multiyear ice, the uncertainty introduced by using salinity profiles as a function of depth is lower than the uncertainty introduced by assuming that the salinity is constant throughout depth (2.3±2.6 K in cold conditions and 28.7±36.5 K in warm conditions). We therefore recommend using an ice salinity profile as a function of depth rather than a constant salinity profile as a simplification.

### 4.2.3 Combining simplified temperature and salinity profiles

In the experiments SIMPLETEMP, SIMPLESALCONST and SIMPLESALFUNC, we learned about the individual effects of using simple temperature and salinity profiles on the brightness temperature simulation. To confirm the conclusion that using both a linear temperature profile and a salinity profile as a function of depth will lead to realistic brightness temperatures, we conduct two additional experiments, combining our simplifications. In the experiment SIMPLEALLCONST, we combine linear temperature profile and constant salinity profile. In the experiment SIMPLEALLFUNC, we combine linear temperature profile and salinity profile as a function of depth.

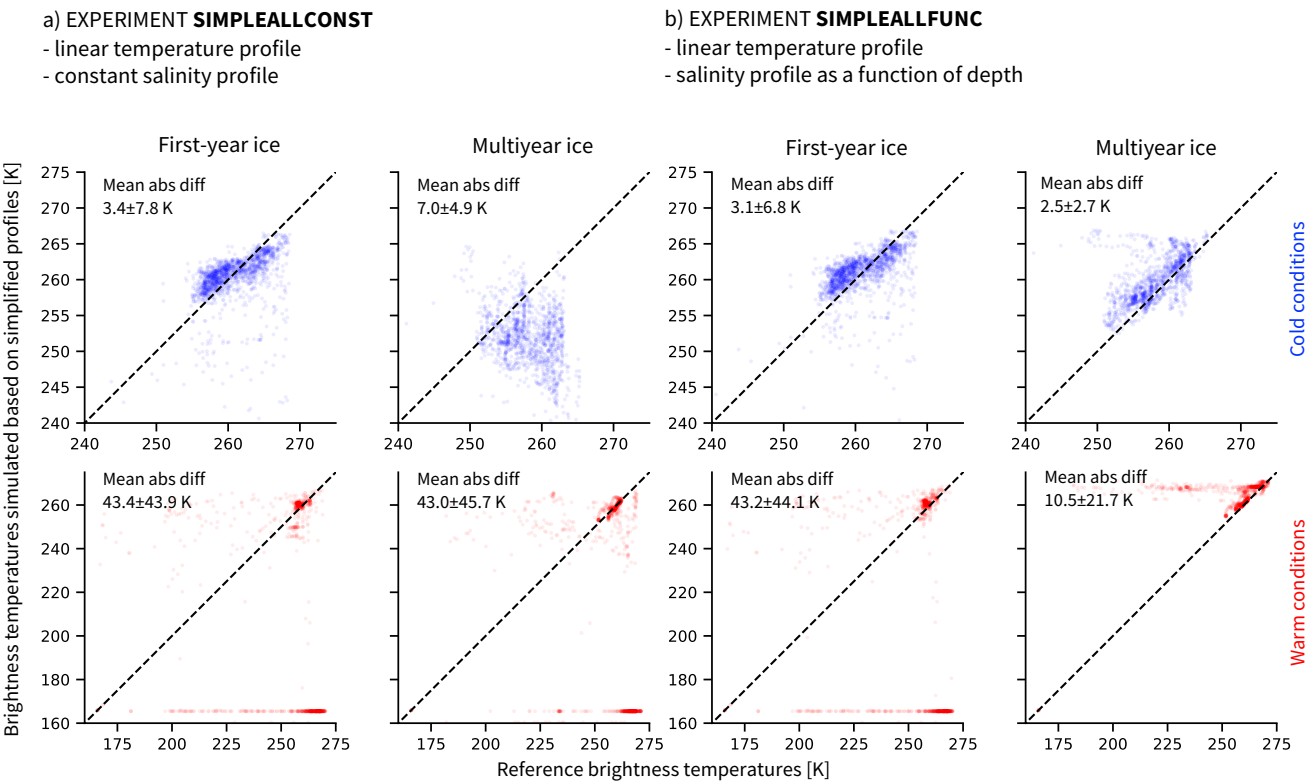

**Figure 7.** Brightness temperatures at 6.9 GHz, vertical polarization, simulated based on two-step linear temperature profiles and (a) constant salinity profiles (experiment SIMPLEALLCONST) or (b) salinity profiles as a function of depth (experiment SIMPLEALLFUNC) as a function of reference brightness temperatures. 1st and 3rd column: first-year ice, 2nd and 4th column column: multiyear ice. Top row: Cold conditions (October to March), bottom row: Warm conditions (April to September). Note that the axes for cold conditions are limited to the range between 240 to 275 K for clarity. The remaining brightness temperatures are scattered between 165 and 240 K and represent around 2% of the simplified data and 0.4% of the reference data.

The results confirm the findings from the previous experiments. In cold conditions, the combination of simplified temperature and salinity profiles leads to brightness temperatures close to reference brightness temperatures for first-year ice, with the

set of profiles using the salinity as a function of depth introducing slightly less uncertainty (3.1±6.8 K) than the set using constant salinity (3.4±7.8 K) (Fig. 7, 1st and 3rd column column). For multiyear ice, the mean absolute difference between the brightness temperatures simulated based on the simplifications and the reference brightness temperatures is clearly lower when using profiles with a salinity as function of depth (2.5±2.7 K) than when using constant salinity profiles (7.0±4.9 K).

In warm conditions, mean absolute differences are one order of magnitude higher than in cold conditions, and similar for first-year ice with both salinity assumptions (43.4±43.9 K using constant salinity and 43.0±44.1 K using salinity as a function of depth). For multiyear ice, again, the uncertainty is clearly lower when using profiles with a salinity as function of depth (10.5±21.7 K) than when using constant salinity profiles (43.0±45.7 K) (Fig. 7, 2nd and 4th column). Reference brightness temperatures and brightness temperatures simulated based on simplified profiles remain clearly different in warm conditions.

Especially, the brightness temperature based on simplified profiles is close or equal to 160 K, i.e. open water brightness temperatures, at most of the time steps in warm conditions. This is because, in warm conditions, the physical temperature of the ice surface approaches 0 °C and, the closer it gets to 0 °C, the lower the salinity must be in order for dry ice to exist rather than slush. At high temperatures and salinities above 0 g/kg (both salinity simplifications for first-year ice, constant salinity for multiyear ice, see Fig. 5), the subsurface brine volume fraction therefore approaches 1 very fast. At subsurface brine volume

fractions near 1, the brightness temperatures approaches the brightness temperature of open water, as shown in Sec. 4.1.

Through these experiments, we investigated the effect of the simplification of temperature and salinity profiles on the simulated brightness temperature. A summary of the setup and results of the different experiments can be found in Table 2. As a conclusion, we recommend using a two-step linear temperature profile in snow and ice and an ice salinity profile as a function of depth when simulating brightness temperatures based on GCM output for cold conditions. For warmer and wet subsurface

conditions, we recommend exploring possibilities to describe surface and subsurface properties as good as possible because the ice subsurface brine volume fraction is the main driver of the simulated brightness temperature.

The effect of temperature and salinity distribution being clearer now, we turn to another characteristic of GCMs, the limited vertical resolution owing to computational efficiency. Indeed, computing vertical temperature and salinity profiles based on the surface temperature and sea-ice thickness given by a GCM adds a vertical dimension to a two-dimensional output. This means

that the computation time and power needed by an operator applied to a GCM will be much higher than a one-dimensional setup. We therefore investigate the importance of the vertical resolution in a next step.

### 4.3    Reducing the vertical resolution

Applying an emission model to a GCM consumes high computation power, as the input profiles must be prepared and the emission model would have to be applied to many grid cells. In the case of the Arctic Ocean at the MPI-ESM low atmospheric

resolution of 1.9°, this would mean for example ≈ 4000 data points per timestep. As ocean components in GCMs often have higher horizontal resolution than the atmosphere, this would mean even more computation power needed when using oceanic variables. Reducing the number of layers for the brightness temperature simulation is a possible aspect to reduce the computation time. This is the issue we explore in the following.

**Table 2.** Summary of the results of the experiments investigating the effect of simplifying temperature and salinity profiles on the brightness temperature simulation. See Sec. 4.2 for more information.

| Experiment | Profiles used | Tools used for profiles | Mean absolute difference between reference and simplified brightness temperatures [K] | | | |
|---|---|---|---|---|---|---|
| | | | Cold Conditions | | Warm conditions | |
| | | | FYI | MYI | FYI | MYI |
| SIMPLETEMP | Linear temperature <br><br><br> Reference salinity | surface temperature as computed by SAMSIM and Eq. A6 as computed by SAMSIM | 2.2±5.8 | 1.2±1.3 | 14.9±23.8 | 10.7±21.3 |
| SIMPLESALCONST | Reference temperature Constant salinity | as computed by SAMSIM 5 g/kg for FYI 1 g/kg for MYI | 2.5±6.6 | 6.6±4.3 | 43.0±44.4 | 40.6±45.5 |
| SIMPLESALFUNC | Reference temperature Salinity as function of depth | as computed by SAMSIM see Table B1 | 2.4±5.9 | 2.3±2.6 | 43.0±44.1 | 28.7±36.5 |
| SIMPLEALLCONST | Linear temperature Constant salinity | see SIMPLETEMP see SIMPLESALCONST | 3.4±7.8 | 7.0±4.9 | 43.4±43.9 | 43.0±45.7 |
| SIMPLEALLFUNC | Linear temperature Salinity as function of depth | see SIMPLETEMP see SIMPLESALFUNC | 3.1±6.8 | 2.5±2.7 | 43.2±44.1 | 10.5±21.7 |

The simplified profiles used for sensitivity experiments in Sec. 4.2 are interpolated to the same number of layers as the reference profiles, i.e. a variable number of layers depending on the ice thickness between one and 100 layers. We now run the brightness temperature simulation with the recommended simplified profile, i.e. linear temperature and salinity as a function of depth, interpolated on ten, seven, five, and three equidistant layers and compare the results to the reference brightness temperatures. We also include the brightness temperatures from the experiment SIMPLEALLFUNC, which is interpolated to the same number of layers as the reference profiles, as an indicator for the minimal simplified uncertainty in the comparison. We concentrate on cold conditions (October to March), as we showed that the uncertainty in warm conditions is already very large at high vertical resolution and mainly depends directly on the upper centimeters rather than on properties further inside the ice.

We find that the difference in uncertainty remains small between the reference simplification between 1 and 100 layers and the interpolation on ten, seven, or five layers, the mean uncertainty staying constant at 3.1 K for first-year ice and varying between 2.4 and 2.5 K for multiyear ice (see Tab. 3). Using three layers, the uncertainty increases slightly by 0.2 K for the former and by 0.8 K for the latter but still remains small. We therefore argue that using as few as five layers is as reasonable as using 100 layers for the simulation of simplified brightness temperatures.

**Table 3.** Absolute mean difference and standard deviation [K] between simplified brightness temperatures simulated based on profiles interpolated on different number of layers and reference brightness temperatures simulated based on profiles covering 1 to 100 layers, depending on the thickness of the ice. These values only represent cold conditions (October to March).

|  | 3 layers | 5 layers | 7 layers | 10 layers | 1 to 100 layers |
|---|---|---|---|---|---|
| First-year ice | 3.3±6.9 | 3.1±6.8 | 3.1±6.8 | 3.1±6.8 | 3.1±6.8 |
| Multiyear ice | 3.3±2.7 | 2.4±2.7 | 2.4±2.7 | 2.4±2.7 | 2.5±2.7 |

## 5 Summary and discussion

### 5.1 Brightness temperatures for cold conditions

We showed that in cold conditions (October to March), we can reproduce realistic sea-ice surface brightness temperatures at 6.9 GHz, vertical polarization, using a two-step linear temperature profile in ice and snow and an ice salinity as a function of depth as input for an emission model. The remaining uncertainty is mainly driven by the simplification of the sea-ice salinity distribution. These realistic brightness temperatures can be reproduced with similar uncertainty using as few as five layers. A very high vertical resolution of the ice properties is therefore not needed.

This study was motivated by the fact that observational uncertainty could be reduced by the approach of an observational operator. It is however not trivial to evaluate this proposition based on our results. To compare the uncertainty [K] introduced by the brightness temperature simulation to uncertainties [%] introduced by a sea-ice concentration retrieval algorithm, we translate the uncertainty in brightness temperature into uncertainty in sea-ice concentration.

A simple retrieval algorithm to retrieve sea-ice concentration $SIC$ is given by

$$\text{SIC} = \frac{\text{TB} - \text{TB}_w}{\text{TB}_i - \text{TB}_w},\tag{1}$$

with TB the total brightness temperature (ice and open water combined), $\text{TB}_w$ a typical open water brightness temperature, and $\text{TB}_i$ a typical sea-ice brightness temperature. If we introduce uncertainties $\Delta\text{SIC}$ and $\Delta\text{TB}$ in the previous equation, this leads to

$$\text{SIC} + \Delta\text{SIC} = \frac{\text{TB} + \Delta\text{TB} - \text{TB}_w}{\text{TB}_i - \text{TB}_w},\tag{2}$$

resulting in

$$\Delta\text{SIC} = \frac{\Delta\text{TB}}{\text{TB}_i - \text{TB}_w}.\tag{3}$$

In our study, we only simulated brightness temperatures of the snow and ice column. To infer an example for $\text{TB}_i$ and $\text{TB}_w$ from our results, we use our finding from Sec. 4.1 that the simulated brightness temperature for ice with low subsurface brine volume fraction is representative for a dry snow and ice column and the simulated brightness temperature for ice with very high subsurface brine volume fraction is comparable to the brightness temperature for open water. From these results we can

therefore infer a $TB_i$, here the simulated brightness temperature for ice with low subsurface brine volume fraction, varying around 263 K (263.8±3.6 K for first-year ice and 263.7±4.3 K for multiyear ice) and a $TB_w$, here the simulated brightness temperatures at very high subsurface brine volume fractions, varying around 166 K (166.1±0.7 K for first-year ice and 165.9±0.1 K for multiyear ice). Following Eq. 3, in this range spanning approximately 100 K, an uncertainty of 1 K in brightness temperature at 6.9 GHz, vertical polarization, therefore approximately translates into 1% of absolute uncertainty in sea-ice concentration. The observational uncertainty of sea-ice concentration in cold conditions is up to 2.5% in consolidated ice and up to 12% for marginal ice zones (Ivanova et al., 2015). The uncertainty of the simulated brightness temperatures translates to a similar range. This might, at first glance, not appear as a solution to drastically reduce the observational uncertainty. However, an observational operator is consistent in time and space and therefore allows a process-understanding of the uncertainties in brightness temperature simulations and, in a possible next step, in retrieval algorithms.

## 5.2 Brightness temperatures for warm conditions

In warm conditions (April to September), we cannot reproduce realistic sea-ice surface brightness temperatures due to the very high sensitivity of the subsurface brine volume fraction to small changes in salinity near 0 °C. We therefore recommend using another approach to simulate brightness temperatures for warm conditions. We suggest assuming that the brightness temperature of warm bare ice is similar all over the Arctic, as temperatures are near 0 °C. The surface brightness temperature is a linear combination of the bare ice brightness temperature and the brightness temperature of the melt ponds covering the ice. Therefore, this constant brightness temperature can be combined with open water brightness temperature, weighted by the fraction of melt ponds forming throughout the warm months. This approach is simple. We have however not found any other approach that could come closer to reality as the sensitivities are very high near 0 °C.

Another problematic component when surface temperatures increase towards warm conditions is the snow. While the detailed profile of dry snow is not necessarily needed as long as its presence is taken into account for the thermal insulation of the ice and for the refraction of the radiation, the influence of wet snow on microwave radiation is much larger. Because in the case of melting snow very precise information about the wetness distribution in the snow is needed, we cannot come close to simulate realistic brightness temperatures from GCM output. In our experiments we have ignored this effect by setting the snow wetness to zero at all times. However, for an all-year-round realistic simulation of brightness temperatures, we suggest to exclude data containing melting snow from the brightness temperature simulation. As periods of wet snow due to melting or advection of warm air, are typically locally limited in time, we argue that our suggestions enable the simulation of brightness temperature simulations over a large amount of the year.

## 5.3 Outlook

The evaluation framework in this study can be used to explore simulated brightness temperatures at higher frequencies, nearer to the most used operational frequencies. However, snow is a limiting factor in this case. While the radiative effect of a dry snow cover is small at 6.9 GHz, its impact increases with increasing frequency. It becomes therefore more important to know the snow structure, e.g. snow density, snow temperature, and snow scatterer structure. This information is lacking in GCMs.

As the snow structure is more dynamic and changes faster than the ice structure, parametrization for the snow structure do not exist yet to our knowledge. It would be of high interest to explore the evolution of snow on sea ice in more details and perform sensitivity studies to identify possible simplifications. These could eventually lead to realistic brightness temperatures simulated based on GCM output at higher frequencies than 6.9 GHz.

Finally, our analysis focuses on the simulation of brightness temperatures based on output from a GCM which simulates sea ice with a very simple sea-ice model. The use of output from GCMs that simulate sea ice with more complex sea-ice models might yield lower uncertainty in the brightness temperature simulation. However, although these models compute many physical properties inside the ice, they do not necessarily store them for each time step. Using the more complex properties of these models would therefore require one to build the emission model into the model code, instead of applying an "external" operator to already produced model output.

## 6   Conclusions

With the help of a one-dimensional thermodynamic sea-ice model and a one-dimensional emission model, we investigated if realistic sea-ice brightness temperatures can be simulated based on GCM output at a frequency of 6.9 GHz with vertical polarization. We conclude that it is possible to simulate realistic sea-ice brightness temperatures if the time of year and boundary conditions are taken into account. We propose the following structure for an observational operator for sea ice at 6.9 GHz, vertical polarization:

*Periods of cold conditions*

- Use the temperature profile provided by the GCM if existing. Otherwise, use the simulated snow surface temperature and oceant temperature at the bottom of the ice to infer a two-step linear temperature profile through the snow and ice.

- Use the salinity profile provided by the GCM if existing. Otherwise, interpolate the salinity profile as a function of depth, following the functions given by Griewank and Notz (2015).

- Apply an emission model, e.g. MEMLS, to these profiles, combined with information about correlation length, sea-ice type, etc.

- Use sea-ice concentration and atmospheric properties provided by the GCM.

- Apply a simple ocean emission model and atmospheric radiative transfer model to account for the influence of open water when the sea-ice concentration is below 100% and for the influence of the atmosphere on the brightness temperature measurements by satellites from space.

*Periods of bare ice near 0 °C*

- Use a constant brightness temperature for the ice surfaces. Burgard et al. (2020) derive a warm conditions sea-ice surface brightness temperature of 266.78 K from observational estimates. This represents a brightness temperature at the top of the atmosphere of 262.29 K corrected by the mean atmospheric effect of 4.49 K in their simulations.

- Use sea-ice concentration, melt pond fraction, and atmospheric properties provided by the GCM.

- Apply a simple ocean emission model and atmospheric radiative transfer model to account for the influence of open water when the sea-ice concentration is below 100% or when melt ponds are present on the ice and for the influence of the atmosphere on the brightness temperature measurements by satellites from space. If not existing yet, include a routine accounting for the effect of melt ponds additionally to the effect of open ocean surfaces in the surface emission model.

*Periods of melting snow*

- Identify periods and locations of reduction in snow thickness at temperatures near 0 °C in the GCM output.

- Ignore these points in the analysis. The GCM output does not provide enough information about the snow properties and wet snow strongly affects the brightness temperature.

The observational operator structure we present here allows us to simulate brightness temperatures from two-dimensional 15 output by a GCM that can be compared with brightness temperatures measured by satellites. This opens new possibilities and perspectives for model-to-observation comparison in the Arctic Ocean.

*Code and data availability.* Primary data and scripts used in this study are archived by the Max Planck Institute for Meteorology and can be obtained by contacting publications@mpimet.mpg.de.

## Appendix A: Retrieving sea-ice properties from temperature and salinity

The following formulas were used to compute the ice density $\rho_i$ and brine volume fraction $\Phi_l$ profiles from the ice temperature $T$ and salinity $S$ profiles:

$$\rho_0 = 916.18 - 0.1403T \tag{A1}$$

where $\rho_0$ is the density of pure ice (Pounder, 1965).

$$S_b = \begin{cases} 508.18 + 14.535T + 0.2018T^2 & \text{if } T \in [-43.2°\text{C}, -36.8°\text{C}] \text{ - Eq. (39) in Vant et al. (1978)} \\ 242.94 + 1.5299T + 0.04529T^2 & \text{if } T \in [-36.8°\text{C}, -22.9°\text{C}] \text{ - Eq. (39) in Vant et al. (1978)} \\ -1.20 - 21.8T - 0.919T^2 & \text{if } T \in ]-22.9°\text{C}, -8.0°\text{C}[ \text{ - Eq. (3.4) in Notz (2005)} \\ 1/(0.001 - (0.05411/T)) & \text{if } T \in [-8.0° \text{ C}, 0°\text{C}[ \text{ - Eq. (3.5) in Notz (2005)} \\ 0 & \text{if } T = 0 \end{cases} \tag{A2}$$

where $S_b$ is the brine salinity.

$$\rho_w = 1000.3 + 0.78237 S_b + 2.8008 \cdot 10^{-4} S_b^2 \qquad \text{(A3)}$$

where $\rho_w$ is the density of seawater at $0°C$ (Eq. (3.8) in Notz, 2005).

$$\Phi_l = \begin{cases} S/S_b & \text{if } S_b > 0 \text{ - Eq. (1.5) in Notz (2005)} \\ 1 & \text{if } S_b = 0 \end{cases} \qquad \text{(A4)}$$

$$\rho_i = \Phi_l \cdot \rho_w + (1 - \Phi_l) \cdot \rho_0 \qquad \text{(A5)}$$

The following formula was used to infer the ice surface temperature $T_{\text{ice,surf}}$ from the snow surface temperature $T_{\text{snow,surf}}$:

10 $$T_{\text{ice,surf}} = \frac{T_{\text{snow,surf}} \cdot \frac{k_s}{h_s} + T_{\text{bottom}} \cdot \frac{k_i}{h_i}}{\frac{k_s}{h_s} + \frac{k_i}{h_i}} \qquad \text{(A6)}$$

with $k_s$ the thermal conductivity of snow (= $0.31 \ W K^{-1} m^{-1}$), $k_i$ the thermal conductivity of ice (= $2.17 \ W K^{-1} m^{-1}$), $h_s$ the snow thickness, $h_i$ the ice thickness, $T_{\text{bottom}}$ the temperature at the bottom of the ice, set to -1.8 °C.

## Appendix B: Salinity parametrization as a function of depth

**Table B1.** Formulas describing salinity as a function of depth, from Griewank and Notz (2015), as shown in the full lines in Fig. 5.

| Ice type | Salinity parametrization as a function of depth $z$ | Constants needed |
|---|---|---|
| First-year ice $S_{\text{fy}}$ | $\frac{z}{a+bz} + c$ | a = 1.0964, b = -1.0552, c = 4.41272 |
| Multiyear ice $S_{\text{my}}$ | $\frac{z}{a} + \left(\frac{z}{b}\right)^{1/c}$ | a = 0.17083, b = 0.92762, c = 0.024516 |
| Transition first-year to multiyear ice | $(1-t) * S_{\text{my}}(z) + t * S_{\text{fy}}(z)$ | t=0 at start of melt season and t=1 at start of freezing season |

*Author contributions.* C.B. carried out all analyses and wrote the manuscript. D.N. and L.T.P. developed the original idea. All authors contributed to discussions.

*Competing interests.* No competing interests are present.

*Acknowledgements.* We thank Stefan Kern for constructive comments and discussions. We also thank Mohammed Shokr and an anonymous
5  reviewer for the detailed and constructive comments. This work was funded by the project "ESA CCI Sea Ice Phase 2".

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
