# Peer review of "The Arctic Ocean Observation Operator for 6.9 GHz (ARC3O) - Part 1: How to obtain sea-ice brightness temperatures at 6.9 GHz from climate model output"

_The Cryosphere, 2019_

## Referee Comment (RC1) · Mohammed Shokr (Referee) · 14 Feb 2020

Review of manuscript "The Arctic Ocean Observation Operator for 6.9 GHz (ARC3O) - Part 1: How to obtain sea-ice brightness temperatures at 6.9 GHz from climate model output"

The manuscript addresses the simulation of brightness temperature at 6.9 GHz from sea ice (with no consideration of snow cover) using ice property profile (temperature and salinity) resulting from an advanced 1D thermodynamic model and other simplified

models. The brightness temperature is simulated using 1D microwave emission model. The main purpose of the study is to examine the sensitivity of the calculated brightness temperature to assumptions of the ice property profiles. With that, the study reached conclusions about the factors that affect the brightness temperature the most, such as the sub-surface salinity of first-year ice and the use of a salinity profile that changes with depth compared to assumption of constant salinity or linear temperature.

With this information, it is possible to develop an observation operator that can be applied to sea ice simulation by a climate model. While the study offers information towards this purpose it does not actually provide a conclusive answer. Yet, this does not take from the credibility of the study as I see it a pioneering attempt to handle the sensitivity issue using a novel approach of testing the effects of different profile shapes (e.g. constant salinity introduced very large uncertainties in brightness temperature and the two-step linear temperature assumption in snow-covered sea ice does not introduce large uncertainties, etc.).

The manuscript is well written and the subject is timely as the issue of sensitivity of ice parameter estimation (from satellite observations or modeling) has been identified as urgent, a conclusion from a workshop on same subject in Hamburg in October 2017.

I would recommend publication subject to revision that takes into consideration the following comments.

Major comments First comment: The writing in some parts is confusing. I had to read the same part several times to understand and connect what the authors want to say. Please modify to make the presentation more coherent, especially in the parts that describe the tested profiles (reference and simplified), sections and sub-section titles that do not reflect the contents, etc. Second comment: The use of some terminology is confusing such as "water liquid volume fraction" Third comment: Some presented aspects of sea ice physics are not precise. I am suggesting corrections.

All these issues are explained in the following comments. I call it minor though they are

many and some may exceed the definition of "minor".

Minor comments: Abstract The last sentence "As periods of melting snow with intermediate moisture content typically last for less than a month, ..." needs modification. Snow may become wet during transition seasons (fall and spring) and that leads to anomalous brightness temperature (please see Shokr et al. Rem Sensing of Env, 123, (2013), and Ye et al., IEEE TGRS, 54(5) (2016)).

Theoretical Background P3 L9: suggest using "loss" instead of "permittivity"

P3 L11: This paragraph is about the emissivity (emitted radiation) from snow-covered sea ice. It needs modifications as I find confusion between using the terms emissivity and permittivity. Here is some information that might be useful in rephrasing the sentences. (1) permittivity determines the reflection/absorption at a surface of dielectric mismatch but emissivity determines the emitted radiation (in TIR or MW bands). (2) While there is relation between the emissivity and reflectivity, there is no relation between emissivity and permittivity. (3) The sentence "This means that water is a stronger absorber than pure ice in the microwave range" is not correct because water has high permittivity as you mentioned in the first sentence, therefore it is high reflector in the MW bands (but not in the TIR). (4) When the snow becomes wet or the ice surface is flooded, the emissivity increases due to the more absorption of solar radiation by water contents (nothing to do with the permittivity). So, to conclude this point, the authors can just focus on the emissivity in this paragraph and remove all connections to the permittivity. Emissivity and permittivity are used in modeling MW emission when layers are assumed (water/ice/snow/air) but this is not the subject of the paragraph.

P3 L16: the sentence "In snow, liquid water is mainly present during melting periods" needs correction. Please see my comment about the Abstract.

P3 L21: the opening of this paragraph "The scattering of the microwave radiation in sea ice is a function of ..." Once again, the theme here should be the emitted radiation, hence the focus should be placed on the two forms of extinction, the absorption and the

scattering. Also, since you include the atmosphere, it is better to mention "the satellite observation of microwave radiation from sea ice" in the first sentence.

P3 L26: you can add "air bubbles in MYI" and mention something about the MW wavelength in relation to the typical size of brine pockets, snow grain, air bubbles and atmospheric droplets.

Method and data P4 L21 This last sentence in the paragraph is clumsy. Please clarify and simplify. P4 L13 Make it "our reference profiles".

P5 L1-5: Any reason why you did not use ERA5?

P5 L11. You provide example to show that simulated sea-ice evolution is not necessarily representative for the real sea-ice evolution at location 75°N, 00°W. You can mention another example at 90°N as this location may not have MYI in all years. Please see maps of MYI in Fig. 8 in Ye et al. (2016) (mentioned above). The maps were generated from a retrieval method using satellite microwave observations.

P6 Fig. 2 the difference between the black and grey lines is not obvious although it is easy to understand what each color indicates. The peak of the ice thickness in June is NOT a "comfortable" result. Equally "uncomfortable" is the rate of MYI thickness increase. My expectation is that MYI thickness increase should take place at a slower rate.

P7 L6: Do you mean "incoming longwave radiation" instead "microwave radiation"?

P7 L7: Table 1, not Tab. 1

P7 L10: just to complete the physics picture, you may add the loss and scattering (extinction) caused by snow wetness, brine wicking, and snow metamorphism. Then you can state that you ignored these effects (the 6.9 GHz is not affected by the grain metamorphism as mentioned before in the text)

P7 L15: it is good to mention this limitation on the application of your study. Just want to

remind you, once again, of the possibility of the wet snow during the transition seasons as indicated above.

P7 Table1: did you mention the source of these data? If not please do.

P7 last 3 lines (no line numbers in the manuscript): this is the first time you mention "brine pocket form". I am not familiar with MEMLS but does it need the geometry of brine pocket? This parameter is not mentioned in Table1.

The influence of vertical sea ice properties Should this section be called "Results"??

P8 second paragraph …. Here are a few observations that might be used to improve the text. First, the salinity profile is always of C-shape as long as the cold temperature prevails. There is a physical explanation. It changes when the temperature rises in the spring. You can refer to the book of Weeks (2015) "On Sea Ice" or the book you already quoted by Shokr and Sinha. Second, the shape of brine pockets does not depend on age but, as rightly stated, on the initial formation process of sea ice. The assumption of spherical pockets may be valid for frazil ice. This is common in the subsurface layer of Antarctic ice and it exists in the Arctic when ice is formed under turbulent oceanic conditions.

P8 Section 4 and section 4.1. The titles do not reflect the contents. For example, Section 4 "The influence of vertical sea ice properties" include Fig. 3, which is about effect of sub-surface salinity (not vertical profile). Also, Section 4.1 "Brine volume fraction" has information about the temperature profile at the end. Please re-organize the information to make improve the flow of the information.

P8 in Section 4.1, the authors kept mentioning "ice surface brine volume" while they mean sub-surface. Please replace "surface" with "sub-surface" and define the subsurface depth, at least roughly.

P9 L1: the sentence "Especially above an ice surface brine volume fraction of 0.2, …" is awkward. You may say "when ice surface brine volume fraction is higher than 0.2

....". Also, it is not right to say "brightness temperature at the ice surface". Just say "brightness temperature from the ice cover". Then, in the following sentence you can say that the radiation is mainly coming from the surface.

P9 L4: in the sentence "brightness temperature transitions roughly linearly .." you may change the word "transitions" to "varies".

P9 L8: the sentence "In our SAMSIM profiles, these high surface brine volume fractions fractions occur predominantly in summer, i.e. from April to September" is correct although the word "fractions" is repeated. I would like to draw the authors' attention to an estimation of brine volume fraction which we performed (experimentally) on Arctic sea ice and found that the fraction in the sub-surface layer (top 5 cm) exceeds 0.2 only when the average temperature exceeds -3°C. It is possible that the temperature of this layer reaches this value in the beginning of the freezing season. But this note does not affect the work in your study.

P9: Figure 3 and the conclusions from this figure are interesting.

P9 last paragraph (no line number) ... you talk about "surface liquid water fraction" and "ice surface brine volume fraction". It is a bit confusing. On P8 L28 you mention "liquid water in the form of brine", which is a bit ambiguous. Brine is brine! And the dissolved salt (not water) is the material that causes loss of MW signal. I would suggest avoiding liquid water and just keep "brine". The liquid water fraction is relevant only to the snow at the onset of melt. Related to this point, you mentioned "For surface liquid water fractions below 0.2, occurring in both winter and summer ..." But Fig. 3 shows surface brine volume fraction, NOT liquid fraction. Also, you said "For these low ice surface brine volume fractions, ...". What are those low fractions? Please fix this issue of liquid water versus brine volume fraction. It is only brine. Not liquid water.

P10 L1-5: it is mentioned that brightness temperature of thin MYI in summer drops to about 180K and that is attributed to the saline layer at the bottom of the ice. It is true that MYI thickens (grows) when winter returns and there is a layer of saline FYI

at the bottom. But why do you say the emitted radiation mainly comes from this layer? The entire volume of the ice radiates. And the radiation from the bottom layer may be completely scattered by the bubbles, which concentrate at the to 20 cm or so.

P10 L6 "Unfortunately for the higher brightness temperatures around 260 K at low ice surface brine volume fractions, we could not infer …..". Are you going back to the FYI here? You are in the middle of discussing MYI.

P10 L9: Again, you mention "liquid water fraction profile". You probably mean brine fraction. Saline FYI ice has slid ice, brine, air and sometimes solid salt if temperature drops below the precipitation point of the salt. MYI has only solid ice and air. The term liquid water fraction is confusing for me.

P10 L1: brightness temperature from MYI is around 180K in winter (low value because of the scattering from air bubbles) and it increases in summer due to surface flooding. That is why you found higher values of 260K. Please correct this information.

P10 L10-14: The information in this paragraph should be combined with information in the first paragraph in section 4.2 (Fig. 4). The current text is confusing. What is the simplified profile? Constant for salinity and linear for temperature? Then why do you include a non-linear salinity in Fig. 4 and call it also "simplified"? Also, MPI-ESM uses the constant salinity and temperature profile. True? Is that the reason you tested the effect of constant salinity on brightness temperature? This is the most confusing part for me. Please re-write to make the information more organized and coherent.

P10 L21: The title of 4.2 does not express the contents. We find data from Reference salinity, Reference temperature and Salinity as function of depth. Also, I would suggest presenting all these options in a table that shows the values, the functions (if any) and the method for each option. That will make it easier for the reader to follow the text and interpret the figure better.

P10 L 22: "as would be given . . ." or better be "as would be used . . ."?
P12 and P13: in the captions of Fig. 5 and Fig. 6 you should mention the season of the data (Oct.-March) and (April-Sept.), respectively.

P14 L8-9: This is the first time the explanation of the non-linear profile in Fig. 4 is explained. That is what I mean by re-organizing the information. I was wondered about this curve while reading, until I reached the explanation here.

P14 Section 4.3: This section highlights the contribution from this study. Would be it useful to compile the statistics of absolute difference in one table to help the reader to explore the impact of each assumption at a glance? The numbers in the text should remain. I am not sure if this suggestion is reasonable but the authors might consider it. The results from using salinity as a function of depth in the case of MYI in summer (Fig. 6) is not the best, contrary to the conclusion in P14 L20.

P14 L28: model or module?

P16 L2: "relationship only depends on the snow thickness". Why depend on snow thickness? You present the decrease of brightness temperature per unit depth (cm)?

P17 L21: "In summer, we cannot reproduce realistic sea-ice surface brightness temperatures due to the very high sensitivity of the liquid water fraction to small changes in salinity near 0°C." Something is wrong here. Brine volume fraction is sensitive to salinity, but liquid water fraction?

P17 L25: the sensitivity of brightness temperature in summer is high because it is related to two parameters which we have no accurate information about; the areal ratio of melt pond and the wetness of the snow or even ice surface as you indicated later. In the next paragraph you mention snow grain as a possible contributor to the brightness temperature in summer. But this influence virtually does not exist at that time.

P18: The Outlook section is well composed. It is true that there is lack of comprehensive data on snow property profiles. However, there are many measurements conducted in scattered areas over the past few decades to characterize snow over ice

under different atmospheric temperatures. It would be useful if someone compiles this information in one review paper and conclude some gross features that can be used in GCM models.

P18 In the Conclusion section there is no mention about the good use of "salinity as a function of depth".

---

## Referee Comment (RC2) · Anonymous Referee #2 · 23 Mar 2020

Summary

The authors consider the development of an observation operator to provide passive microwave brightness data at 6.9 GHz frequency and Vertical polarization. The work is motivated by the need to overcome observational uncertainty introduced by geophysical retrieval algorithms applied to satellite observations and used to initialize and evaluate climate models. Here, the observation operator simulates the brightness temperature from the climate model output instead of requiring the retrieved sea ice concentration from observed brightness temperature data. Consideration of the feasibility

and limitations of the observation operator concept for simulated sea ice is the main focus here. The authors use highly resolved 1D thermodynamic sea-ice and 1D microwave emission models to consider the effect that the simplified temperature and salinity profiles characteristic of GCM outputs have on brightness temperature estimates and observation operator performance. Generally, the approach works well for cold, winter conditions, and in the peak of summer when surface melt ponds are present, but not during periods of wet snow. The authors determine the boundary conditions for the construction of an operator that is evaluated against satellite brightness temperatures in their companion paper (which I did not evaluate).

In general the paper is well written and the descriptions and figures are mostly clear and concise. Appendix A is useful for providing equations though Appendix B is just a table that could be in the paper. The methods should be better organized, and made to be distinct from the results, to make the paper easier to follow. For example, on Page 10, around line 11, there are new methods and their reasoning described in amongst the section focused on the results presented in Figure 3.

The authors should clarify their positioning on the role that snow plays on the examined 6.9 GHz frequency and vertical polarization, in the contexts of season, ice type, and other available frequencies and polarizations. It is mostly all there, just hard to follow. For example, the negligible contribution of dry snow properties compared to ice (due to brine in the ice) is cited is advantageous for the $\sim$4.3 cm wavelength examined, yet there is a section looking into the role of dry snow (Section 6) and the following statement is made "the radiative effect of the snow cover hence remains important.". At the beginning of Section 7.3 snow is cited as a limiting factor. Perhaps it is better to make it clearer earlier in the paper that one of the goals of the study is to better understand the potential impact of dry (and wet snow) conditions on the operator output. Statements about wet snow are easier to follow as there are not contradictions.

Detailed Comments

P = Page; L=Line

P3L22: 'atmosphere' doesn't fit here because the sentence is referring to sea ice.

P5L7-10: The purpose behind defining specific locations is unclear. This is especially true since the authors indicate that sea ice seldom exists at the first-year sea ice location. The choice of locations for the sensitivity analysis are also arbitrary. If the choice of location does not affect the study then the locational context isn't needed.

P7: The paragraph on the bottom, beginning "Our input for the emission model…", is somewhat dismissive of the breadth of in-situ observations that are available, and the role of these observations in model development. It would be clearer is the authors outlined the model set-up, inputs, and assumptions used, since this is a methods section, and save uncertainty evaluations and suggestions for the discussion section.

P9L8: Is it correct to say that April in the Arctic is summer?

P9 Figure 3: Symbols for FYI and MYI are not clear in the figure.

P14L3: It is confusing that the assumption of constant salinity introduces large uncertainties in the brightness temperature during summer, when earlier the authors mentioned the properties inside the ice do not influence the brightness temperature when the ice surface has a brine volume fraction higher than 0.2 (also during summer). Also on P15 (L7-8) the authors say the brightness temperature depends on the surface rather than internal ice properties. Some clarification given in the context of expected penetration depth would be helpful.

P16L15-17: Indicate what would happen if ice concentration were <100%.

P16L18: Section 7 should be "Discussion and Conclusion".

P17L19-20: Sentence "In summer…" is confusing i.e. how is the liquid water fraction highly sensitive to changes in salinity. Do you mean the salinity of the melt ponds?

P18L24: correct "temperatur"

P18L33: The authors should elaborate on how the brightness temperature would be weighted by melt pond fraction.

P19L3: How would periods of wet snow be identified?

P19 Appendix A: Indicate the validity ranges of the formulas.

---

## Author Comment (AC1) · 30 Apr 2020

RC: Reviewer comment, AR: Author response

Reviewer Summary: The manuscript addresses the simulation of brightness temperature at 6.9 GHz from sea ice (with no consideration of snow cover) using ice property profile (temperature and salinity) resulting from an advanced 1D thermodynamic model and other simplified models. The brightness temperature is simulated using 1D microwave emission model. The main purpose of the study is to examine the sensitivity

of the calculated brightness temperature to assumptions of the ice property profiles. With that, the study reached conclusions about the factors that affect the brightness temperature the most, such as the sub-surface salinity of first-year ice and the use of a salinity profile that changes with depth compared to assumption of constant salinity or linear temperature. With this information, it is possible to develop an observation operator that can be applied to sea ice simulation by a climate model. While the study offers information towards this purpose it does not actually provide a conclusive answer. Yet, this does not take from the credibility of the study as I see it a pioneering attempt to handle the sensitivity issue using a novel approach of testing the effects of different profile shapes (e.g. constant salinity introduced very large uncertainties in brightness temperature and the two-step linear temperature assumption in snow-covered sea ice does not introduce large uncertainties, etc.). The manuscript is well written and the subject is timely as the issue of sensitivity of ice parameter estimation (from satellite observations or modeling) has been identified as urgent, a conclusion from a workshop on same subject in Hamburg in October 2017. I would recommend publication subject to revision that takes into consideration the following comments.

AR: Thank you very much for the positive feedback, and for your detailed, constructive comments on how to further improve our paper. We plan to address all your comments as described in the following.

—

RC: Major comments First comment: The writing in some parts is confusing. I had to read the same part several times to understand and connect what the authors want to say. Please modify to make the presentation more coherent, especially in the parts that describe the tested profiles (reference and simplified), sections and sub-section titles that do not reflect the contents, etc.

AR: We will clarify the structure, better separating the methods and the results. We also plan to split some of the figures into several figures to make it easier for the reader

to follow.

—

RC: Second comment: The use of some terminology is confusing such as "water liquid volume fraction".

AR: In the process of working on this study, we have moved from using the term "liquid water fraction" to using the term "brine volume fraction". We started with "liquid water fraction" in opposition to "solid ice fraction" but decided to move on with "brine volume fraction" to avoid the confusion you are mentioning. Therefore, there should not have been mentions of "liquid water fraction" left in the manuscript. We apologize for the confusion and will replace "liquid water" by "brine" in the relevant occurrences.

—

RC: Third comment: Some presented aspects of sea ice physics are not precise. I am suggesting corrections.

AR: Thank you for the suggestions given below. We will take them into account in the revision of the manuscript.

—

RC: Minor comments (All these issues are explained in the following comments. I call it minor though they are many and some may exceed the definition of "minor".) Abstract The last sentence "As periods of melting snow with intermediate moisture content typically last for less than a month,..." needs modification. Snow may become wet during transition seasons (fall and spring) and that leads to anomalous brightness temperature (please see Shokr et al. Rem Sensing of Env, 123,(2013), and Ye et al., IEEE TGRS, 54(5) (2016)).

AR: We will clarify based on your suggestion.

—
[Figure]

RC: P3 L9: suggest using "loss" instead of "permittivity"

AR: Done.

—

RC: P3 L11: This paragraph is about the emissivity (emitted radiation) from snow-covered sea ice. It needs modifications as I find confusion between using the terms emissivity and permittivity. Here is some information that might be useful in rephrasing the sentences. (1) permittivity determines the reflection/absorption at a surface of dielectric mismatch but emissivity determines the emitted radiation (in TIR or MW bands). (2) While there is relation between the emissivity and reflectivity, there is no relation between emissivity and permittivity. (3) The sentence "This means that water is a stronger absorber than pure ice in the microwave range" is not correct because water has high permittivity as you mentioned in the first sentence, therefore it is high reflector in the MW bands (but not in the TIR). (4) When the snow becomes wet or the ice surface is flooded, the emissivity increases due to the more absorption of solar radiation by water contents (nothing to do with the permittivity). So, to conclude this point, the authors can just focus on the emissivity in this paragraph and remove all connections to the permittivity. Emissivity and permittivity are used in modeling MW emission when layers are assumed (water/ice/snow/air) but this is not the subject of the paragraph.

AR: We apologize for the confusion due to using "permittivity" instead of "emissivity". Thank you for the clarification. We will change the terminology according to your suggestion.

—

RC: P3 L16: the sentence "In snow, liquid water is mainly present during melting periods" needs correction. Please see my comment about the Abstract.

AR: We will clarify following your suggestion above.

RC: P3 L21: the opening of this paragraph "The scattering of the microwave radiation in sea ice is a function of..." Once again, the theme here should be the emitted radiation, hence the focus should be placed on the two forms of extinction, the absorption and the scattering. Also, since you include the atmosphere, it is better to mention "the satellite observation of microwave radiation from sea ice" in the first sentence.

AR: Thank you for pointing out that this sentence was not precise. We will clarify following your suggestion.

—

RC: P3 L26: you can add "air bubbles in MYI" and mention something about the MW wavelength in relation to the typical size of brine pockets, snow grain, air bubbles and atmospheric droplets.

AR: We will add the air bubbles and typical sizes of the scattering bodies into the text.

—

RC: P4 L21 This last sentence in the paragraph is clumsy. Please clarify and simplify.

AR: Thank you for pointing that out. We will reformulate this sentence.

—

RC: P4 L13 Make it "our reference profiles".

AR: Done.

—

RC: P5 L1-5: Any reason why you did not use ERA5?

AR: Most of the analysis presented here was conducted and finished before the release of ERA5. However, we do not expect the choice of reanalysis data to substantially affect

the results of the study in any case, as the analysis focuses on conceptual findings, not tied to the exact timing and location of the forcing.

—

RC: P5 L11. You provide example to show that simulated sea-ice evolution is not necessarily representative for the real sea-ice evolution at location 75°N, 00°W. You can mention another example at 90°N as this location may not have MYI in all years. Please see maps of MYI in Fig. 8 in Ye et al. (2016) (mentioned above). The maps were generated from a retrieval method using satellite microwave observations.

AR: As mentioned in the manuscript, we do not claim to simulate the sea-ice evolution at the given location and time realistically. This is because SAMSIM always assumes a seasonal cycle for the oceanic heat flux to the bottom of the ice following the oceanic heat flux measured during the SHEBA campaign north of Alaska. Under the combination of ERA-Interim atmospheric forcing and this SHEBA oceanic forcing, sea ice can form at 75N00W and the ice at the North Pole survives the summer melt. This also means that locations which usually have MYI as pointed out in the reference you give might not have MYI in our simulations. As suggested by reviewer #2, we will explain the principle and location more conceptually. This will highlight that the locations for which the ERA-Interim forcing was chosen cannot be compared to these locations in reality.

—

RC: P6 Fig.2 the difference between the black and grey lines is not obvious although it is easy to understand what each color indicates. The peak of the ice thickness in June is NOT a "comfortable" result. Equally "uncomfortable" is the rate of MYI thickness increase. My expectation is that MYI thickness increase should take place at a slower rate.

AR: As mentioned in the caption, the peak of ice thickness in June is a model artifact.

As they represent only a very small fraction of data points, we do not expect this to have an effect on our results. However, to avoid confusion, we will mask these points out for the study. To our knowledge, the MYI thickness increase is not anomalous. Following your remark, we will check with literature again.

—

RC: P7 L6: Do you mean "incoming longwave radiation" instead "microwave radiation"?

AR: We mean microwave radiation. This is the radiation normally referred to as the downwelling microwave radiation. This represents all microwave radiation reaching the ground from the atmosphere. Contributors to this radiation are background space radiation, clouds and water vapour in the atmosphere, and oxygen. However, we set it to 0 K in our setup because we are mainly interested in the effects of sea-ice physical properties on the brightness temperature. We will clarify this in the manuscript.

—

RC: P7 L7: Table 1, not Tab. 1

AR: Changed.

—

RC: P7 L10: just to complete the physics picture, you may add the loss and scattering (extinction) caused by snow wetness, brine wicking, and snow metamorphism. Then you can state that you ignored these effects (the 6.9 GHz is not affected by the grain metamorphism as mentioned before in the text)

AR: We will complete the sentence.

—

RC: P7 L15: it is good to mention this limitation on the application of your study. Just want to remind you, once again, of the possibility of the wet snow during the transition

seasons as indicated above.

AR: We will clarify the handling of dry and wet snow throughout the text.

—

RC: P7 Table1: did you mention the source of these data? If not please do.

AR: We will provide the sources for these constants in the caption.

—

RC: P7 last 3 lines (no line numbers in the manuscript): this is the first time you mention "brine pocket form". I am not familiar with MEMLS but does it need the geometry of brine pocket? This parameter is not mentioned in Table 1.

AR: MEMLS assumes either random needles or spherical pockets. We mention later in the manuscript that we use the spherical pockets assumption. Following your suggestion, we will include this information earlier.

—

RC: The influence of vertical sea ice properties Should this section be called "Results"??

AR: This will be part of restructuring the manuscript.

—

RC: P8 second paragraph.... Here are a few observations that might be used to improve the text. First, the salinity profile is always of C-shape as long as the cold temperature prevails. There is a physical explanation. It changes when the temperature rises in the spring. You can refer to the book of Weeks (2015) "On Sea Ice" or the book you already quoted by Shokr and Sinha. Second, the shape of brine pockets does not depend on age but, as rightly stated, on the initial formation process of sea ice. The assumption of spherical pockets may be valid for frazil ice. This is common in
the subsurface layer of Antarctic ice and it exists in the Arctic when ice is formed under turbulent oceanic conditions.

AR: Thank you, this is useful information. We will clarify following your input.

—

RC: P8 Section 4 and section 4.1. The titles do not reflect the contents. For example, Section 4 "The influence of vertical sea ice properties" include Fig. 3, which is about effect of sub-surface salinity (not vertical profile). Also, Section 4.1 "Brine volume fraction" has information about the temperature profile at the end. Please re-organize the information to make improve the flow of the information.

AR: Thank you for your input. This will be part of restructuring the manuscript.

—

RC: P8 in Section 4.1, the authors kept mentioning "ice surface brine volume" while they mean sub-surface. Please replace "surface" with "sub-surface" and define the subsurface depth, at least roughly.

AR: We apologize for the confusion. In this case our subsurface is the upper 1 centimeter. We will clarify by using "near-surface brine volume".

—

RC: P9 L1: the sentence "Especially above an ice surface brine volume fraction of 0.2,..." is awkward. You may say "when ice surface brine volume fraction is higher than 0.2 ...". Also, it is not right to say "brightness temperature at the ice surface". Just say "brightness temperature from the ice cover". Then, in the following sentence you can say that the radiation is mainly coming from the surface.

AR: We will change the sentence following your suggestion.

—
RC: P9 L4: in the sentence "brightness temperature transitions roughly linearly .." you may change the word "transitions" to "varies".

AR: Done.

—

RC: P9 L8: the sentence "In our SAMSIM profiles, these high surface brine volume fractions fractions occur predominantly in summer, i.e. from April to September" is correct although the word "fractions" is repeated. I would like to draw the authors' attention to an estimation of brine volume fraction which we performed (experimentally) on Arctic sea ice and found that the fraction in the sub-surface layer (top 5 cm) exceeds 0.2 only when the average temperature exceeds -3°C. It is possible that the temperature of this layer reaches this value in the beginning of the freezing season. But this note does not affect the work in your study.

AR: Thank you for pointing this repetition out. We think that your observations are in line with our findings. Thanks for sharing these!

—

RC: P9: Figure 3 and the conclusions from this figure are interesting.

AR: We agree, thank you.

—

RC: P9 last paragraph (no line number)... you talk about "surface liquid water fraction" and "ice surface brine volume fraction". It is a bit confusing. On P8 L28 you mention "liquid water in the form of brine", which is a bit ambiguous. Brine is brine! And the dissolved salt (not water) is the material that causes loss of MW signal. I would suggest avoiding liquid water and just keep "brine". The liquid water fraction is relevant only to the snow at the onset of melt. Related to this point, you mentioned "For surface liquid water fractions below 0.2, occurring in both winter and summer..." But Fig.3 shows

surface brine volume fraction, NOT liquid fraction. Also, you said "For these low ice surface brine volume fractions,...". What are those low fractions? Please fix this issue of liquid water versus brine volume fraction. It is only brine. Not liquid water.

AR: Again, we apologize for the confusion. As mentioned in an answer to a previous comment, in the process of working on this study, we have moved from using the term "liquid water fraction" to using the term "brine volume fraction". We started with "liquid water fraction" in opposition to "solid ice fraction" but decided to move on with "brine volume fraction" to avoid the confusion you are mentioning. Therefore, there should not have been mentions of "liquid water fraction" left in the manuscript. We apologize for the confusion and will replace "liquid water" by "brine" in the relevant occurrences.

—

RC: P10 L1-5: it is mentioned that brightness temperature of thin MYI in summer drops to about 180K and that is attributed to the saline layer at the bottom of the ice. It is true that MYI thickens (grows) when winter returns and there is a layer of saline FYI at the bottom. But why do you say the emitted radiation mainly comes from this layer? The entire volume of the ice radiates. And the radiation from the bottom layer may be completely scattered by the bubbles, which concentrate at the to 20 cm or so.

AR: The influence of the bottom salinity on the MYI brightness temperature was inferred from investigating the different profiles one by one. These low MYI brightness temperatures were only found in a few September and October occurrences. In these cases, the ice is not thicker than 20 cm and the only property that can explain this difference when looking at the data is the gradient in salinity in the bottom layer. We therefore assume that the penetration depth reaches the ocean below the ice in these cases. As the ice thickens again during the freezing period, this effect vanishes rapidly. AS such thin MYI is not very common in the Arctic, we do not expect this issue to be relevant when inferring brightness temperatures from actual climate model output.

—

RC: P10 L6 "Unfortunately for the higher brightness temperatures around 260 K at low ice surface brine volume fractions, we could not infer...". Are you going back to the FYI here? You are in the middle of discussing MYI.

AR: The structure will be revisited following the general change in structure.

—

RC: P10 L9: Again, you mention "liquid water fraction profile". You probably mean brine fraction. Saline FYI ice has slid ice, brine, air and sometimes solid salt if temperature drops below the precipitation point of the salt. MYI has only solid ice and air. The term liquid water fraction is confusing for me.

AR: Again, we apologize for the confusion. This will be corrected.

—

RC: P10 L1: brightness temperature from MYI is around 180K in winter (low value because of the scattering from air bubbles) and it increases in summer due to surface flooding. That is why you found higher values of 260K. Please correct this information.

AR: This is not the case here. Our high values around 250 K are what is expected at 6.9 GHz. Typical tie-points values for winter MYI lie near 250 K (e.g. Ivanova et al. 2015, TC Vol9(5) use 246K). Low brightness temperatures for MYI are only occurring in our simulation in rare occasions during September and October when the MYI is at minimum thickness, but the surface is not wet anymore. As explained in a previous comment, these low brightness temperatures are anomalies tied to thin MYI, which does not occur often in the Arctic.

—

RC: P10 L10-14: The information in this paragraph should be combined with information in the first paragraph in section 4.2 (Fig.4). The current text is confusing. What is the simplified profile? Constant for salinity and linear for temperature? Then why do

you include a non-linear salinity in Fig. 4 and call it also "simplified"? Also, MPI-ESM uses the constant salinity and temperature profile. True? Is that the reason you tested the effect of constant salinity on brightness temperature? This is the most confusing part for me. Please re-write to make the information more organized and coherent.

AR: Again, we will work on a new structure to clarify.

—

RC: P10 L21: The title of 4.2 does not express the contents. We find data from Reference salinity, Reference temperature and Salinity as function of depth. Also, I would suggest presenting all these options in a table that shows the values, the functions (if any) and the method for each option. That will make it easier for the reader to follow the text and interpret the figure better.

AR: Again, we will work on a new structure to clarify.

—

RC: P10 L22: "as would be given..." or better be "as would be used..."?

AR: Replaced.

—

RC: P12 and P13: in the captions of Fig.5 and Fig.6 you should mention the season of the data (Oct.-March) and (April-Sept.), respectively.

AR: Thank you for pointing that out. We will add the clarification.

—

RC: P14 L8-9: This is the first time the explanation of the non-linear profile in Fig. 4 is explained. That is what I mean by re-organizing the information. I was wondered about this curve while reading, until I reached the explanation here.

AR: Again, this will be part of the restructuring of the manuscript.

—

RC: P14 Section 4.3: This section highlights the contribution from this study. Would be it useful to compile the statistics of absolute difference in one table to help the reader to explore the impact of each assumption at a glance? The numbers in the text should remain. I am not sure if this suggestion is reasonable but the authors might consider it. The results from using salinity as a function of depth in the case of MYI in summer (Fig. 6) is not the best, contrary to the conclusion in P14 L20.

AR: Yes, the salinity as a function of depth leads to the best result for MYI in warm conditions (10.6+/-21.7 K compared to 43.0+/-45.7 K for constant salinity). We will try your suggestion of using a table. This might be an important way to conclude these results, especially if we split the figures.

—

RC: P14 L28: model or module?

AR: We mean "model" here. We do not plan to integrate the emission model as a module into the climate model but rather to apply it on already produced climate model output.

—

RC: P16 L2: "relationship only depends on the snow thickness". Why depend on snow thickness? You present the decrease of brightness temperature per unit depth (cm)?

AR: We plan on looking into the data again and clarify.

—

RC: P17 L21: "In summer, we cannot reproduce realistic sea-ice surface brightness temperatures due to the very high sensitivity of the liquid water fraction to small changes in salinity near 0°C." Something is wrong here. Brine volume fraction is sensitive to salinity, but liquid water fraction?

[Figure]

AR: Again, we apologize for the confusion. We mean "brine volume fraction" and will replace it.

—

RC: P17 L25: the sensitivity of brightness temperature in summer is high because it is related to two parameters which we have no accurate information about; the areal ratio of melt pond and the wetness of the snow or even ice surface as you indicated later. In the next paragraph you mention snow grain as a possible contributor to the brightness temperature in summer. But this influence virtually does not exist at that time.

AR: We agree, this is unclear. We will clarify this.

—

RC: P18: The Outlook section is well composed. It is true that there is lack of comprehensive data on snow property profiles. However, there are many measurements conducted in scattered areas over the past few decades to characterize snow over ice under different atmospheric temperatures. It would be useful if someone compiles this information in one review paper and conclude some gross features that can be used in GCM models.

AR: Yes, we strongly agree that such a compilation of observations would be a very valuable resource for similar studies in the future.

—

RC: P18 In the Conclusion section there is no mention about the good use of "salinity as a function of depth".

AR: We have mentioned the salinity as a function of depth in the point about "cold conditions". We will work on highlighting this better.

---

## Author Comment (AC2) · 30 Apr 2020

RC: Reviewer Comment, AR: Author Response

RC: Reviewer summary: The authors consider the development of an observation operator to provide passive microwave brightness data at 6.9 GHz frequency and Vertical polarization. The work is motivated by the need to overcome observational uncertainty introduced by geophysical retrieval algorithms applied to satellite observations and used to initialize and evaluate climate models. Here, the observation operator

simulates the brightness temperature from the climate model output instead of requiring the retrieved sea ice concentration from observed brightness temperature data. Consideration of the feasibility and limitations of the observation operator concept for simulated sea ice is the main focus here. The authors use highly resolved 1D thermodynamic sea-ice and 1D microwave emission models to consider the effect that the simplified temperature and salinity profiles characteristic of GCM outputs have on brightness temperature estimates and observation operator performance. Generally, the approach works well for cold, winter conditions, and in the peak of summer when surface melt ponds are present, but not during periods of wet snow. The authors determine the boundary conditions for the construction of an operator that is evaluated against satellite brightness temperatures in their companion paper (which I did not evaluate). In general the paper is well written and the descriptions and figures are mostly clear and concise. Appendix A is useful for providing equations though Appendix B is just a table that could be in the paper. The methods should be better organized, and made to be distinct from the results, to make the paper easier to follow. For example, on Page 10, around line 11, there are new methods and their reasoning described in amongst the section focused on the results presented in Figure 3. The authors should clarify their positioning on the role that snow plays on the examined 6.9 GHz frequency and vertical polarization, in the contexts of season, ice type, and other available frequencies and polarizations. It is mostly all there, just hard to follow. For example, the negligible contribution of dry snow properties compared to ice (due to brine in the ice) is cited is advantageous for the $\sim$4.3 cm wavelength examined, yet there is a section looking into the role of dry snow (Section 6) and the following statement is made "the radiative effect of the snow cover hence remains important.". At the beginning of Section 7.3 snow is cited as a limiting factor. Perhaps it is better to make it clearer earlier in the paper that one of the goals of the study is to better understand the potential impact of dry (and wet snow) conditions on the operator output. Statements about wet snow are easier to follow as there are not contradictions.

AR: Thank you very much for the positive feedback, and for your detailed, constructive

comments on how to further improve our paper. We will work on a new clearer structure for the manuscript. Also we plan to clarify the issue of snow for our study further. We plan to address the other comments as described in the following.

—

RC: P3L22: 'atmosphere' doesn't fit here because the sentence is referring to sea ice.

AR: Thank you for pointing that out. We will reformulate the sentence to clarify that we are describing the brightness temperature measured by the satellite from space.

—

RC: P5L7-10: The purpose behind defining specific locations is unclear. This is especially true since the authors indicate that sea ice seldom exists at the first-year sea ice location. The choice of locations for the sensitivity analysis are also arbitrary. If the choice of location does not affect the study then the locational context isn't needed.

AR: We will follow your suggestion and try to describe the forcing data in a more conceptual way.

—

RC: P7: The paragraph on the bottom, beginning "Our input for the emission model...", is somewhat dismissive of the breadth of in-situ observations that are available, and the role of these observations in model development. It would be clearer is the authors outlined the model set-up, inputs, and assumptions used, since this is a methods section, and save uncertainty evaluations and suggestions for the discussion section.

AR: We agree that it is more common to discuss uncertainties after presenting the results. However, in this case, we want to make clear to the reader right in the beginning that, while there might be many uncertainties, they do not affect our results substantially. This way, the reader can concentrate on our results without being concerned about these limitations while reading the paper.
—

RC: P9L8: Is it correct to say that April in the Arctic is summer?

AR: We apologize for the confusion. To be more precise, we will change "summer" to "warm conditions" and "winter" to "cold conditions".

—

RC: P9 Figure 3: Symbols for FYI and MYI are not clear in the figure.

AR: Thank you for pointing that out. We will work on another for differences between FYI and MYI.

—

RC: P14L3: It is confusing that the assumption of constant salinity introduces large uncertainties in the brightness temperature during summer, when earlier the authors mentioned the properties inside the ice do not influence the brightness temperature when the ice surface has a brine volume fraction higher than 0.2 (also during summer). Also on P15 (L7-8) the authors say the brightness temperature depends on the surface rather than internal ice properties. Some clarification given in the context of expected penetration depth would be helpful.

AR: We apologize for the confusion. We will clarify this.

—

RC: P16L15-17: Indicate what would happen if ice concentration were <100 %.

AR: Here, we check the effect of the atmosphere on the brightness temperature over a sea-ice surface. The radiative transfer model we use to compute the atmospheric influence on the brightness temperature also includes the computation of the ocean surface brightness temperature. There is no possibility of separating the two and using the atmospheric module alone. As we are solely interested in the brightness temperature

above ice surfaces in this study, we therefore have to assume 100 % sea-ice concentration as input to this radiative transfer model. Using an ice concentration smaller than 100 % would mean having the effect of the ocean surface on the brightness temperature as well.

—

RC: P16L18: Section 7 should be "Discussion and Conclusion".

AR: We acknowledge that this would be a more typical way of structuring the manuscript. However, we prefer to keep Section 8 as a short conclusion and leave Section 7 to a Summary and Discussion. However, we will rethink Section 7 in the process of restructuring the manuscript.

—

RC: P17L19-20: Sentence "In summer..." is confusing i.e. how is the liquid water fraction highly sensitive to changes in salinity. Do you mean the salinity of the melt ponds?

AR: We apologize for the use of "liquid water fraction" here, we actually mean "brine volume fraction". We will reformulate to avoid confusion. The brine volume fraction is highly sensitive to changes in bulk salinity and temperature. As temperatures are near 0°C, ice can only exist at very low salinities. The brine volume fraction increases very fast for low brine salinities but salinities of 1 g/kg or even more, following Eq. A4.

—

RC: P18L33: The authors should elaborate on how the brightness temperature would be weighted by melt pond fraction.

AR: We will clarify our suggestion for this process.

—

RC: P19L3: How would periods of wet snow be identified?

AR: Here also, we will clarify our suggestion for this process.

—

RC: P19 Appendix A: Indicate the validity ranges of the formulas.

AR: We are sorry if this is not clear. We will add the validity ranges.

---

## Author Response (AR1)

**Final Author Comments**

**The Arctic Ocean Observation Operator for 6.9 GHz (ARC3O) - Part 1: How to obtain sea-ice brightness temperatures at 6.9 GHz from climate model output**

Burgard, C., Notz, D., Pedersen, L.T., Tonboe, R.T.
*The Cryosphere,* #10.5194/tc-2019-317
* * *
**RC: Reviewer Comment**,     AR: Author Response,     *changed manuscript text*

AR:  We thank both reviewers for taking the time to read through our paper with such detailed attention. We acknowledge the concerns about the structure and have focused on restructuring the manuscript more clearly. Also, we are very grateful for all the more localized but precise remarks and suggestions. We hope to have fulfilled your expectations and to have clarified your concerns. Larger changes in the manuscript include:

- A new structure for the discussion of the results, presenting the list of experiments before presenting the results, separated figures for the experiments, and a summarizing table

- Removing the section about snow and atmosphere as they were confusing and not fitting the scope of the study, which is to assess the effect of using simple GCM ice and snow information as input for the brightness temperature simulation

**1. Reviewer #1**

RC:  **Reviewer Summary:**
**The manuscript addresses the simulation of brightness temperature at 6.9 GHz from sea ice (with no consideration of snow cover) using ice property profile (temperature and salinity) resulting from an advanced 1D thermodynamic model and other simplified models. The brightness temperature is simulated using 1D microwave emission model. The main purpose of the study is to examine the sensitivity of the calculated brightness temperature to assumptions of the ice property profiles. With that, the study reached conclusions about the factors that affect the brightness temperature the most, such as the sub-surface salinity of first-year ice and the use of a salinity profile that changes with depth compared to assumption of constant salinity or linear temperature.**
**With this information, it is possible to develop an observation operator that can be**

**applied to sea ice simulation by a climate model. While the study offers information towards this purpose it does not actually provide a conclusive answer. Yet, this does not take from the credibility of the study as I see it a pioneering attempt to handle the sensitivity issue using a novel approach of testing the effects of different profile shapes (e.g. constant salinity introduced very large uncertainties in brightness temperature and the two-step linear temperature assumption in snow-covered sea ice does not introduce large uncertainties, etc.).**

**The manuscript is well written and the subject is timely as the issue of sensitivity of ice parameter estimation (from satellite observations or modeling) has been identified as urgent, a conclusion from a workshop on same subject in Hamburg in October 2017.**

**I would recommend publication subject to revision that takes into consideration the following comments.**

AR:  Thank you very much for the positive feedback, and for your detailed, constructive comments on how to further improve our paper. We have addressed all your comments as described in the following.

RC:  *Major comments*
**First comment: The writing in some parts is confusing. I had to read the same part several times to understand and connect what the authors want to say. Please modify to make the presentation more coherent, especially in the parts that describe the tested profiles (reference and simplified), sections and sub-section titles that do not reflect the contents, etc.**

AR:  We have tried to clarify the structure, better separating the methods and the results. We also have splitted some of the figures into several figures to make it easier for the reader to follow.
*See Sec. 3.4 and Sec. 4*

RC:  **Second comment: The use of some terminology is confusing such as "water liquid volume fraction".**

AR:  In the process of working on this study, we have moved from using the term "liquid water fraction" to using the term "brine volume fraction". We started with "liquid water fraction" in opposition to "solid ice fraction" but decided to move on with "brine volume fraction" to avoid the confusion you are mentioning. Therefore, there should not have been mentions of "liquid water fraction" left in the manuscript. We apologize for the confusion and have replaced "liquid water" by "brine" in the relevant occurrences.

**RC:** **Third comment: Some presented aspects of sea ice physics are not precise. I am suggesting corrections.**

 AR: Thank you for the suggestions given below. We have taken them into account in the revision of the manuscript.

**RC:** *Minor comments (All these issues are explained in the following comments. I call it minor though they are many and some may exceed the definition of "minor".)***:**
**Abstract The last sentence "As periods of melting snow with intermediate moisture content typically last for less than a month,..." needs modification. Snow may become wet during transition seasons (fall and spring) and that leads to anomalous brightness temperature (please see Shokr et al. Rem Sensing of Env, 123,(2013), and Ye et al., IEEE TGRS, 54(5) (2016)).**

 AR: We have tried to clarify based on your suggestion as follows:
*Finally, in our setup, we cannot assess the effect of wet snow properties. As periods of snow with intermediate moisture content, typically occuring in spring and fall, locally last for less than a month, our approach allows one to estimate realistic brightness temperatures at 6.9 GHz from climate model output for most of the year.*

**RC:** **P3 L9: suggest using "loss" instead of "permittivity"**

 AR: We have reformulated this sentence as follows:
*It depends on the temperature distribution in the medium and on the transmission and reflection affecting the path of the microwave radiation from the emitting layer within the medium to the surface of the medium.*

**RC:** **P3 L11: This paragraph is about the emissivity (emitted radiation) from snow-covered sea ice. It needs modifications as I find confusion between using the terms emissivity and permittivity. Here is some information that might be useful in rephrasing the sentences. (1) permittivity determines the reflection/absorption at a surface of dielectric mismatch but emissivity determines the emitted radiation (in TIR or MW bands). (2) While there is relation between the emissivity and reflectivity, there is no relation between emissivity and permittivity. (3) The sentence "This means that water is a stronger absorber than pure ice in the microwave range" is not correct because water has high permittivity as you mentioned in the first sentence, therefore it is high reflector in the MW bands (but not in the TIR). (4) When the snow becomes wet or the ice surface is flooded, the emissivity increases due to the more absorption of solar radiation by water contents (nothing to do with the permittivity). So, to conclude this point, the authors can just focus on the emissivity in this paragraph and remove all connections to the permittivity. Emissivity**

**and permittivity are used in modeling MW emission when layers are assumed (water/ice/snow/air) but this is not the subject of the paragraph.**

AR: We apologize for the confusion due to using "permittivity" instead of "emissivity". Thank you for the clarification. We have restructured this section to clarify that we are describing the radiative properties of the layered snow and ice column, influencing the resulting brightness temperature and therefore the permittivity is important.
*See Sec. 2*

RC: **P3 L16: the sentence "In snow, liquid water is mainly present during melting periods" needs correction. Please see my comment about the Abstract.**

AR: We have tried to clarify following your suggestion above. We now write:
*If the snow becomes wet, as happens during melting periods and localized events of warm air advection mainly occurring in spring and fall, the dielectric loss in the snow layers increases substantially, leading to a reduction in the transmissivity of the snow layer to microwave radiation.*

RC: **P3 L21: the opening of this paragraph "The scattering of the microwave radiation in sea ice is a function of..." Once again, the theme here should be the emitted radiation, hence the focus should be placed on the two forms of extinction, the absorption and the scattering. Also, since you include the atmosphere, it is better to mention "the satellite observation of microwave radiation from sea ice" in the first sentence.**

AR: Thank you for pointing out that this sentence was not precise. We have restructured Sec. 2 to correct the imprecision.

RC: **P3 L26: you can add "air bubbles in MYI" and mention something about the MW wavelength in relation to the typical size of brine pockets, snow grain, air bubbles and atmospheric droplets.**

AR: We have added the air bubbles and typical sizes of the scattering bodies into the text. We now write:
*In general, scattering affects the brightness temperature measured from space over sea-ice surfaces increasingly with increasing frequency (Tonboe et al., 2006) as the wavelength successively approaches the size of brine pockets and air bubbles on the order of tenths of millimeters to millimeters, snow grains on the order of hundreds of micrometers to millimeters and atmospheric aerosols and droplets on the order of micrometers.*

**RC: P4 L21 This last sentence in the paragraph is clumsy. Please clarify and simplify.**

AR: Thank you for pointing that out. We have reformulated as follows:
*This a necessary first step to understand fundamental drivers of the brightness temperature before comparing brightness temperatures simulated on the basis of MPI-ESM output directly to brightness temperatures measured by satellites, which we do in Burgard et al., 2020.*

**RC: P4 L13 Make it "our reference profiles".**

AR: Done.

**RC: P5 L1-5: Any reason why you did not use ERA5?**

AR: Most of the analysis presented here was conducted and finished before the release of ERA5. However, we do not expect the choice of reanalysis data to substantially affect the results of the study in any case, as the analysis focuses on conceptual findings, not tied to the exact timing and location of the forcing.

**RC: P5 L11. You provide example to show that simulated sea-ice evolution is not necessarily representative for the real sea-ice evolution at location 75°N, 00°W. You can mention another example at 90°N as this location may not have MYI in all years. Please see maps of MYI in Fig. 8 in Ye et al. (2016) (mentioned above). The maps were generated from a retrieval method using satellite microwave observations.**

AR: As mentioned in the manuscript, we do not claim to simulate the sea-ice evolution at the given location and time realistically. This is because SAMSIM always assumes a seasonal cycle for the oceanic heat flux to the bottom of the ice following the oceanic heat flux measured during the SHEBA campaign north of Alaska. Under the combination of ERA-Interim atmospheric forcing and this SHEBA oceanic forcing, sea ice can form at 75N00W and the ice at the North Pole survives the summer melt. This also means that locations which usually have MYI as pointed out in the reference you give might not have MYI in our simulations.
As suggested by reviewer #2, we have tried to explain the principle and location more conceptually. This highlights that the locations for which the ERA-Interim forcing was chosen cannot be compared to these locations in reality. We have included the information about the exact location for reproducibility in the caption of Fig.2. We have reformulated as follows:
*We conduct our analysis using atmospheric forcing from two random points in the Arctic Ocean as input for SAMSIM. At the first point, the combined forcing of the ERA-Interim atmospheric variables and the SHEBA oceanic flux leads to complete melting of the simulated ice in summer each year, resulting in several*

*cycles of first-year ice. At the second point, the combination of the atmospheric forcing and oceanic heat flux leads to a simulated ice cover present throughout the year, resulting in several cycles of multiyear ice (Fig.2). This way, we capture potential differences in the brightness temperature simulation depending on the ice type. To ensure that the conclusions we draw from these two random points are robust, we have conducted the same analysis on five additional random points distributed in the Arctic Ocean and the results support our conclusions.*

**RC:** **P6 Fig.2 the difference between the black and grey lines is not obvious although it is easy to understand what each color indicates. The peak of the ice thickness in June is NOT a "comfortable" result. Equally "uncomfortable" is the rate of MYI thickness increase. My expectation is that MYI thickness increase should take place at a slower rate.**

AR: As mentioned in the caption, the peak of ice thickness in June is a model artifact. As they represent only a very small fraction of data points, we do not expect this to have an effect on our results. However, to avoid confusion, we have now masked these points out for the study. Regarding the rate of increase of MYI, we agree with you. In this case, we are looking at a comparably fast growth because it is comparably thin MYI. Compared to the FYI growth rate in the left panel of Fig.2, the MYI growth rate is slower.

**RC:** **P7 L6: Do you mean "incoming longwave radiation" instead "microwave radiation"?**

AR: We mean microwave radiation. This is the radiation normally referred to as the down-welling microwave radiation. This represents all microwave radiation reaching the ground from the atmosphere. Contributors to this radiation are background space radiation, clouds and water vapour in the atmosphere, and oxygen. However, we set it to 0 K in our setup because we are mainly interested in the effects of sea-ice physical properties on the brightness temperature. We have reformulated as follows:
*These are the correlation length, the brine pocket form, the incidence angle, the ocean temperature, the incoming microwave radiation from the atmosphere (i.e. the cosmic background radiation and the radiation reflected and emitted by properties of the atmosphere) and the ice-ocean reflectivity for vertical polarization.*

**RC:** **P7 L7: Table 1, not Tab. 1**

AR: Changed.

**RC:** **P7 L10: just to complete the physics picture, you may add the loss and scattering (extinction) caused by snow wetness, brine wicking, and snow metamorphism.**

**Then you can state that you ignored these effects (the 6.9 GHz is not affected by the grain metamorphism as mentioned before in the text)**

AR: We have now separated this explanation into dry and wet snow and added you suggestion as follows:
*The effect of wet snow on the brightness temperature is larger and depends on the snow wetness, brine wicking, and snow metamorphism.*

**RC: P7 L15: it is good to mention this limitation on the application of your study. Just want to remind you, once again, of the possibility of the wet snow during the transition seasons as indicated above.**

AR: We have reformulated as follows:
*However, when comparing results of a possible observation operator based on this study to actual observations, we strongly recommend to not consider periods of wet snow, during melting periods and events of warm air advection, as setting the snow wetness to zero will lead to unplausible brightness temperatures in these periods.*

**RC: P7 Table1: did you mention the source of these data? If not please do.**

AR: We have now added the sources for these constants in the caption as follows:
*MEMLS constant input details and properties of the snow layer. The incidence angle is from AMSR-E and AMSR2, passive microwave sensors measuring at 6.9 GHZ (NASDA, 2003; JAXA, 2011). The ocean temperature and snow density are the constant values used in a GCM such as MPI-ESM (Wetzel et al., 2012; Giorgetta et al., 2013). The incoming microwave radiation from the atmosphere is set to 0 K because we want to focus on the effect of sea-ice properties on the emitted radiation. Correlation lengths are based on past experiments conducted by R.T. Tonboe.*

**RC: P7 last 3 lines (no line numbers in the manuscript): this is the first time you mention "brine pocket form". I am not familiar with MEMLS but does it need the geometry of brine pocket? This parameter is not mentioned in Table 1.**

AR: MEMLS assumes either random needles or spherical pockets. We use the spherical pockets assumption but as scattering is negligible at 6.9 GHz, we argue that the choice of brine pocket form will not affect our results substantially. We have added the information about the brine pocket form in Table 1 and added following sentence in the text:
*In any case, we assume that the choice of brine pocket form will not affect our result substantially because scattering within the ice is negligible at 6.9 GHz.*

**RC:** **The influence of vertical sea ice properties Should this section be called "Results"??**

 AR: In the course of restructuring the manuscript, we now call this "Results".

**RC:** **P8 second paragraph.... Here are a few observations that might be used to improve the text. First, the salinity profile is always of C-shape as long as the cold temperature prevails. There is a physical explanation. It changes when the temperature rises in the spring. You can refer to the book of Weeks (2015) "On Sea Ice" or the book you already quoted by Shokr and Sinha. Second, the shape of brine pockets does not depend on age but, as rightly stated, on the initial formation process of sea ice. The assumption of spherical pockets may be valid for frazil ice. This is common in the subsurface layer of Antarctic ice and it exists in the Arctic when ice is formed under turbulent oceanic conditions.**

 AR: Thank you, this is useful information. We have reformulated following your input:
*The salinity parametrization used in Sec. 4.2.2 is based on an "L-shape" of the salinity profile, while the sea-ice salinity profile often resembles a "C-shape" or even a "T-shape" when cold temperatures prevail (Nakawo and Sinha, 1981; Shokr and Sinha, 2015a).*
and:
*However, it is known that the brine pocket form highly depends on the initial formation process of the ice, which is not simulated.*

**RC:** **P8 Section 4 and section 4.1. The titles do not reflect the contents. For example, Section 4 "The influence of vertical sea ice properties" include Fig. 3, which is about effect of sub-surface salinity (not vertical profile). Also, Section 4.1 "Brine volume fraction" has information about the temperature profile at the end. Please re-organize the information to make improve the flow of the information.**

 AR: Thank you for your input. We have taken this comment into account when we restructured the manuscript. We have now called the Section containing Fig.3 "Subsurface properties vs. Vertical profile".

**RC:** **P8 in Section 4.1, the authors kept mentioning "ice surface brine volume" while they mean sub-surface. Please replace "surface" with "sub-surface" and define the subsurface depth, at least roughly.**

 AR: We apologize for the confusion. In this case our subsurface is the upper 1 centimeter. We now follow your suggestion by using the term "ice subsurface brine volume".

**RC:** **P9 L1: the sentence "Especially above an ice surface brine volume fraction of 0.2,..." is awkward. You may say "when ice surface brine volume fraction is higher than 0.2 ...". Also, it is not right to say "brightness temperature at the ice surface". Just say "brightness temperature from the ice cover". Then, in the following sentence you can say that the radiation is mainly coming from the surface.**

 AR: Thank you for the suggestion. We have now changed the sentence as follows:
*When the ice subsurface brine volume fraction is higher than 0.2, the brightness temperature from the ice column is linearly related to the ice subsurface brine volume fraction (Fig. 3, bottom row).*

**RC:** **P9 L4: in the sentence "brightness temperature transitions roughly linearly .." you may change the word "transitions" to "varies".**

 AR: Done.

**RC:** **P9 L8: the sentence "In our SAMSIM profiles, these high surface brine volume fractions fractions occur predominantly in summer, i.e. from April to September" is correct although the word "fractions" is repeated. I would like to draw the authors' attention to an estimation of brine volume fraction which we performed (experimentally) on Arctic sea ice and found that the fraction in the sub-surface layer (top 5 cm) exceeds 0.2 only when the average temperature exceeds -3°C. It is possible that the temperature of this layer reaches this value in the beginning of the freezing season. But this note does not affect the work in your study.**

 AR: Thank you for pointing this repetition out. We think that your observations are in line with our findings. Thanks for sharing these! We have added this information by completing the following sentence with "the beginning of the freezing season":
*In our SAMSIM profiles, these high subsurface brine volume fractions occur predominantly in warm conditions, i.e. from April to September, during the melting period and in the beginning of the freezing season.*

**RC:** **P9: Figure 3 and the conclusions from this figure are interesting.**

 AR: We agree, thank you.

**RC:** **P9 last paragraph (no line number)... you talk about "surface liquid water fraction" and "ice surface brine volume fraction". It is a bit confusing. On P8 L28 you mention "liquid water in the form of brine", which is a bit ambiguous. Brine is brine! And the dissolved salt (not water) is the material that causes loss of MW signal. I would suggest avoiding liquid water and just keep "brine". The liquid water fraction is relevant only to the snow at the onset of melt. Related to this**

**point, you mentioned "For surface liquid water fractions below 0.2, occurring in both winter and summer..." But Fig.3 shows surface brine volume fraction, NOT liquid fraction. Also, you said "For these low ice surface brine volume fractions,...". What are those low fractions? Please fix this issue of liquid water versus brine volume fraction. It is only brine. Not liquid water.**

AR: Again, we apologize for the confusion. As mentioned in an answer to a previous comment, in the process of working on this study, we have moved from using the term "liquid water fraction" to using the term "brine volume fraction". We started with "liquid water fraction" in opposition to "solid ice fraction" but decided to move on with "brine volume fraction" to avoid the confusion you are mentioning. Therefore, there should not have been mentions of "liquid water fraction" left in the manuscript. We apologize for the confusion and have replaced "liquid water" by "brine" in the relevant occurrences.

RC: **P10 L1-5: it is mentioned that brightness temperature of thin MYI in summer drops to about 180K and that is attributed to the saline layer at the bottom of the ice. It is true that MYI thickens (grows) when winter returns and there is a layer of saline FYI at the bottom. But why do you say the emitted radiation mainly comes from this layer? The entire volume of the ice radiates. And the radiation from the bottom layer may be completely scattered by the bubbles, which concentrate at the to 20 cm or so.**

AR: The influence of the bottom salinity on the MYI brightness temperature was inferred from investigating the different input profiles one by one. These low MYI brightness temperatures were found in September, in the first two weeks of the re-freezing period. In the corresponding input profiles, the ice salinity is zero for all layers except in the bottom layer, where new saline ice is forming. As scattering is negligible as 6.9 GHz, the brightness temperature can be influenced by this bottom layer. However, we agree that this might not necessarily be realistic and that the conditions leading to these salinity profiles could be investigated further. As a side note, this phenomenon does not occur anymore when using simplified temperature and salinity profiles. We have reformulated as follows:

*In some multiyear ice cases during warm conditions, the brightness temperature drops below 240 K at near-zero subsurface brine volume fractions. These low brightness temperatures occur in September, in the first two or three weeks in which ice growth sets in again. In these cases, the ice column used as input for MEMLS has a brine volume fraction of zero over the whole column, except in the bottom layer. We therefore suggest that the simulated brightness temperature is mainly influenced by the very saline bottom layer at the interface between ice and ocean in these cases, leading to low brightness temperatures. This behaviour is not necessarily realistic and the conditions leading to these input salinity profiles might need further investigation.*

**RC:** **P10 L6 "Unfortunately for the higher brightness temperatures around 260 K at low ice surface brine volume fractions, we could not infer...". Are you going back to the FYI here? You are in the middle of discussing MYI.**

**AR:** We have now restructured this discussion and hope it is clearer.

**RC:** **P10 L9: Again, you mention "liquid water fraction profile". You probably mean brine fraction. Saline FYI ice has slid ice, brine, air and sometimes solid salt if temperature drops below the precipitation point of the salt. MYI has only solid ice and air. The term liquid water fraction is confusing for me.**

**AR:** Again, we apologize for the confusion. This has been corrected.

**RC:** **P10 L1: brightness temperature from MYI is around 180K in winter (low value because of the scattering from air bubbles) and it increases in summer due to surface flooding. That is why you found higher values of 260K. Please correct this information.**

**AR:** This is not the case here. Our high values around 250 K are what is expected at 6.9 GHz. Typical tie-points values for winter MYI lie near 250 K (e.g. Ivanova et al. 2015, TC Vol9(5) use 246K). Low brightness temperatures for MYI are only occurring in our simulation in rare occasions during September in the beginning of the freezing season.

**RC:** **P10 L10-14: The information in this paragraph should be combined with information in the first paragraph in section 4.2 (Fig.4). The current text is confusing. What is the simplified profile? Constant for salinity and linear for temperature? Then why do you include a non-linear salinity in Fig. 4 and call it also "simplified"? Also, MPI-ESM uses the constant salinity and temperature profile. True? Is that the reason you tested the effect of constant salinity on brightness temperature? This is the most confusing part for me. Please re-write to make the information more organized and coherent.**

**AR:** Again, this comment has been taken into account for the new structure. We hope it is clearer now.

**RC:** **P10 L21: The title of 4.2 does not express the contents. We find data from Reference salinity, Reference temperature and Salinity as function of depth. Also, I would suggest presenting all these options in a table that shows the values, the functions (if any) and the method for each option. That will make it easier for the reader to follow the text and interpret the figure better.**

**AR:** Again, this comment has been taken into account for the new structure. We also have

added Table 2, which includes the information about the experiment setup and the results.

**RC:** **P10 L22: "as would be given..." or better be "as would be used..."?**

AR: Replaced.

**RC:** **P12 and P13: in the captions of Fig.5 and Fig.6 you should mention the season of the data (Oct.-March) and (April-Sept.), respectively.**

AR: Thank you for pointing that out. We have added this clarification.

**RC:** **P14 L8-9: This is the first time the explanation of the non-linear profile in Fig. 4 is explained. That is what I mean by re-organizing the information. I was wondered about this curve while reading, until I reached the explanation here.**

AR: We have re-organized the manuscript so that the two profiles are discussed in the beginning of Sec.4.2.2. We now write the following:
*In the experiment SIMPLESALCONST, we explore the effect of a constant salinity profile on the simulated brightness temperature. MPI-ESM assumes a constant salinity of 5 g/kg regardless of sea-ice type or age. As this is clearly too high for multiyear ice (Ulaby et al., 1986), we assume a constant salinity of 5 g/kg for first-year ice and a constant salinity of 1 g/kg for multiyear ice throughout the ice column in our simplified salinity profiles (see dashed lines in Fig.5).*
*In the parallel experiment SIMPLESALFUNC, we explore an alternative approach to simplify salinity profiles. We use a parametrization representing salinity as a function of depth (Griewank and Notz, 2015). This parametrization assumes an L-shaped profile, with low salinity near the surface and a rapidly increasing salinity in the lower ice layers (see Fig.5, full lines, and Table B1). This parametrization has been evaluated against observations (Griewank and Notz, 2015). In both SIMPLESALCONST and SIMPLESALFUNC, we use the reference temperature profiles simulated by SAMSIM.*

**RC:** **P14 Section 4.3: This section highlights the contribution from this study. Would be it useful to compile the statistics of absolute difference in one table to help the reader to explore the impact of each assumption at a glance? The numbers in the text should remain. I am not sure if this suggestion is reasonable but the authors might consider it. The results from using salinity as a function of depth in the case of MYI in summer (Fig. 6) is not the best, contrary to the conclusion in P14 L20.**

AR: Yes, the salinity as a function of depth, combined with the linear temperature profile, leads to the best result for MYI in warm conditions (10.5±21.7 K compared to 43.0±45.7 K for constant salinity).

We have followed your suggestion and added Table 2 as a summary of the experiment results.

**RC:** **P14 L28: model or module?**

 AR: We mean "model" here. We do not plan to integrate the emission model as a module into the climate model but rather to apply it on already produced climate model output.

**RC:** **P16 L2: "relationship only depends on the snow thickness". Why depend on snow thickness? You present the decrease of brightness temperature per unit depth (cm)?**

 AR: We have removed this section as it was diverting from the main message of the paper, which is the properties of the ice column that are needed. Instead we have added the following paragraph in the initial discussion about potential uncertainties in Sec.3.3:
*Another limitation in the input data for MEMLS is the snow information. We investigated the indirect effect of the snow cover on the simulated brightness temperature, e.g. the radiative effect (as opposed to the thermal insulation effect), and found that the brightness temperature decreases by approximately 0.13 K for every centimeter of snow present on the ice column. Therefore, although the snow is expected to be "transparent" at less than 10 GHz, lack of information about the snow structure besides snow temperature and thickness might still lead to uncertainties of up to a few K in the presence of a thick snow cover.*

**RC:** **P17 L21: "In summer, we cannot reproduce realistic sea-ice surface brightness temperatures due to the very high sensitivity of the liquid water fraction to small changes in salinity near 0°C." Something is wrong here. Brine volume fraction is sensitive to salinity, but liquid water fraction?**

 AR: Again, we apologize for the confusion. We mean "brine volume fraction" and have replaced it.

**RC:** **P17 L25: the sensitivity of brightness temperature in summer is high because it is related to two parameters which we have no accurate information about; the areal ratio of melt pond and the wetness of the snow or even ice surface as you indicated later. In the next paragraph you mention snow grain as a possible contributor to the brightness temperature in summer. But this influence virtually does not exist at that time.**

 AR: We agree, this is unclear. We have removed the mention of the influence of snow grains on the brightness temperature in summer.

**RC:** **P18: The Outlook section is well composed. It is true that there is lack of compre-
hensive data on snow property profiles. However, there are many measurements
conducted in scattered areas over the past few decades to characterize snow over
ice under different atmospheric temperatures. It would be useful if someone com-
piles this information in one review paper and conclude some gross features that
can be used in GCM models.**

**AR:** Yes, we strongly agree that such a compilation of observations would be a very valuable
resource for similar studies in the future.

**RC:** **P18 In the Conclusion section there is no mention about the good use of "salinity
as a function of depth".**

**AR:** We have mentioned the salinity as a function of depth in the point about "cold conditions".
We have restructured the conclusion and hope this is highlighted better now. We now
write:
*Periods of cold conditions*

- *Use the temperature profile provided by the GCM if existing. Otherwise,
  use the simulated snow surface temperature and oceant temperature at
  the bottom of the ice to infer a two-step linear temperature profile through
  the snow and ice.*

- *Use the salinity profile provided by the GCM if existing. Otherwise, interpo-
  late the salinity profile as a function of depth, following the functions given
  by Notz and Griewank, 2015.*

- *Apply an emission model, e.g. MEMLS, to these profiles, combined with
  information about correlation length, sea-ice type, etc.*

- *Use sea-ice concentration, and atmospheric properties provided by the
  GCM.*

- *Apply a simple ocean emission model and atmospheric radiative transfer
  model to account for the influence of open water when the sea-ice con-
  centration is below 100% and for the influence of the atmosphere on the
  brightness temperature measurements by satellites from space.*

**2. Reviewer #2**

RC: **Reviewer summary:**

**The authors consider the development of an observation operator to provide passive microwave brightness data at 6.9 GHz frequency and Vertical polarization. The work is motivated by the need to overcome observational uncertainty introduced by geophysical retrieval algorithms applied to satellite observations and used to initialize and evaluate climate models. Here, the observation operator simulates the brightness temperature from the climate model output instead of requiring the retrieved sea ice concentration from observed brightness temperature data. Consideration of the feasibility and limitations of the observation operator concept for simulated sea ice is the main focus here. The authors use highly resolved 1D thermodynamic sea-ice and 1D microwave emission models to consider the effect that the simplified temperature and salinity profiles characteristic of GCM outputs have on brightness temperature estimates and observation operator performance. Generally, the approach works well for cold, winter conditions, and in the peak of summer when surface melt ponds are present, but not during periods of wet snow. The authors determine the boundary conditions for the construction of an operator that is evaluated against satellite brightness temperatures in their companion paper (which I did not evaluate).**

**In general the paper is well written and the descriptions and figures are mostly clear and concise. Appendix A is useful for providing equations though Appendix B is just a table that could be in the paper. The methods should be better organized, and made to be distinct from the results, to make the paper easier to follow. For example, on Page 10, around line 11, there are new methods and their reasoning described in amongst the section focused on the results presented in Figure 3.**

**The authors should clarify their positioning on the role that snow plays on the examined 6.9 GHz frequency and vertical polarization, in the contexts of season, ice type, and other available frequencies and polarizations. It is mostly all there, just hard to follow. For example, the negligible contribution of dry snow properties compared to ice (due to brine in the ice) is cited is advantageous for the $\approx 4.3$ cm wavelength examined, yet there is a section looking into the role of dry snow (Section 6) and the following statement is made "the radiative effect of the snow cover hence remains important.". At the beginning of Section 7.3 snow is cited as a limiting factor. Perhaps it is better to make it clearer earlier in the paper that one of the goals of the study is to better understand the potential impact of dry (and wet snow) conditions on the operator output. Statements about wet snow are easier to follow as there are not contradictions.**

AR: Thank you very much for the positive feedback, and for your detailed, constructive comments on how to further improve our paper. We have worked on a new structure for the manuscript and have tried to further clarify the issue of snow for our study.

We have addressed your other comments as described in the following.

**RC:    P3L22: 'atmosphere' doesn't fit here because the sentence is referring to sea ice.**

AR:    Thank you for pointing that out. We have reformulated the sentence to clarify that we are describing the brightness temperature measured by the satellite from space. We now write as follows:
*As brightness temperatures are usually not measured at the ice surface but at the top of the atmosphere by satellites, the microwave radiation emitted by the sea-ice cover can additionally be affected by transmissivity and reflectivity of the snow and atmosphere on the path between the surface and the satellite.*

**RC:    P5L7-10: The purpose behind defining specific locations is unclear. This is especially true since the authors indicate that sea ice seldom exists at the first-year sea ice location. The choice of locations for the sensitivity analysis are also arbitrary. If the choice of location does not affect the study then the locational context isn't needed.**

AR:    We have followed your suggestion and now describe the forcing data in a more conceptual way, as follows:
*We conduct our analysis using atmospheric forcing from two random points in the Arctic Ocean as input for SAMSIM. At the first point, the combined forcing of the ERA-Interim atmospheric variables and the SHEBA oceanic flux leads to complete melting of the simulated ice in summer each year, resulting in several cycles of first-year ice. At the second point, the combination of the atmospheric forcing and oceanic heat flux leads to a simulated ice cover present throughout the year, resulting in several cycles of multiyear ice (Fig.2). This way, we capture potential differences in the brightness temperature simulation depending on the ice type. To ensure that the conclusions we draw from these two random points are robust, we have conducted the same analysis on five additional random points distributed in the Arctic Ocean and the results support our conclusions.*

**RC:    P7: The paragraph on the bottom, beginning "Our input for the emission model...", is somewhat dismissive of the breadth of in-situ observations that are available, and the role of these observations in model development. It would be clearer is the authors outlined the model set-up, inputs, and assumptions used, since this is a methods section, and save uncertainty evaluations and suggestions for the discussion section.**

AR:    We agree that it is more common to discuss uncertainties after presenting the results. However, in this case, we want to make clear to the reader right in the beginning that, while there might be many uncertainties, they do not affect our results substantially.

This way, the reader can concentrate on our results without being concerned about these limitations while reading the paper.

**RC:** **P9L8: Is it correct to say that April in the Arctic is summer?**

 AR: We apologize for the confusion. To be more precise, we have changed all occurrences of "summer" to "warm conditions" and "winter" to "cold conditions" throughout the manuscript.

**RC:** **P9 Figure 3: Symbols for FYI and MYI are not clear in the figure.**

 AR: Thank you for pointing that out. We have now divided the results for FYI and MYI in two separate subfigures.
*See Fig.3*

**RC:** **P14L3: It is confusing that the assumption of constant salinity introduces large uncertainties in the brightness temperature during summer, when earlier the authors mentioned the properties inside the ice do not influence the brightness temperature when the ice surface has a brine volume fraction higher than 0.2 (also during summer). Also on P15 (L7-8) the authors say the brightness temperature depends on the surface rather than internal ice properties. Some clarification given in the context of expected penetration depth would be helpful.**

 AR: We apologize for the confusion. Wit a new structure of the manuscript, we hope to have clarified this point. We now use the results of Sec.4.1. to assess in which conditions information about the vertical profile is needed and when not. In many warm conditions cases it is not needed, but there are also warm conditions cases in which the ice subsurface brine volume fraction is below 0.2 and therefore profile information is needed. Also the simplified profiles are also relevant for the subsurface layer (especially for salinity), so this is why we look into the influence of simplified profiles for warm conditions as well.

**RC:** **P16L15-17: Indicate what would happen if ice concentration were <100%.**

 AR: We have removed this section as it was beyond the scope of this study. We focus on the brightness temperature simulated for a snow and ice column, based on profiles that could be inferred from GCM output. We realize that this section was confusing. Discussing the effect of the atmosphere, which can be accounted for by using a radiative transfer model, and regions of less than 100% sea ice are beyond the scope of this study.

**RC:** **P16L18: Section 7 should be "Discussion and Conclusion".**

 AR: We acknowledge that this would be a more typical way of structuring the manuscript.

However, we prefer to keep Section 6 (previously 8) as a short conclusion with the main take-home messages and leave Section 5 (previously 7) to a Summary and Discussion.

**RC:** **P17L19-20: Sentence "In summer..." is confusing i.e. how is the liquid water fraction highly sensitive to changes in salinity. Do you mean the salinity of the melt ponds?**

AR: We apologize for the use of "liquid water fraction" here, we actually mean "brine volume fraction". We have replaced it. We mean the salinity of the subsurface layer. The brine volume fraction is highly sensitive to changes in bulk salinity and temperature. As temperatures are near $0°$C, ice can only exist at very low salinities. The brine volume fraction increases very fast for low brine salinities (A4) but the salinities we use in our simplified profiles are often of 1 g/kg or even more.

**RC:** **P18L33: The authors should elaborate on how the brightness temperature would be weighted by melt pond fraction.**

AR: To weight by melt pond fractions, we suggest using the melt pond fraction given by the GCM and treat it as an open water surface when combining the results of the ocean emission model and our sea-ice brightness temperatures. We have reformulated as follows:
*Periods of bare ice near 0 °C*

- *Use a constant brightness temperature for the ice surfaces. Burgard et al., 2020 derive a warm conditions sea-ice surface brightness temperature of 266.78 K from observational estimates. This represents a brightness temperature at the top of the atmosphere of 262.29 K corrected by the mean atmospheric effect of 4.49 K in their simulations.*

- *Use sea-ice concentration, melt pond fraction, and atmospheric properties provided by the GCM.*

- *Apply a simple ocean emission model and atmospheric radiative transfer model to account for the influence of open water when the sea-ice concentration is below 100% or when melt ponds are present on the ice and for the influence of the atmosphere on the brightness temperature measurements by satellites from space. If not existing yet, include a routine accounting for the effect of melt ponds additionally to the effect of open ocean surfaces in the surface emission model.*

**RC:** **P19L3: How would periods of wet snow be identified?**

AR:  We have tried to clarify as follows in the conclusions:
     *Periods of melting snow*

     - *Identify periods and locations of reduction in snow thickness at temperatures near 0 °C in the GCM output.*

     - *Ignore these points in the analysis. The GCM output does not provide enough information about the snow properties and wet snow strongly affects the brightness temperature.*

**RC:  P19 Appendix A: Indicate the validity ranges of the formulas.**

AR:  We are sorry if this is not clear. We have added the validity ranges and updated outdated formulas.
     *See Appendix A*

[revised manuscript text omitted]